# SARS-CoV-2 disease severity and transmission efficiency is increased for airborne compared to fomite exposure in Syrian hamsters

Julia R. Port[1,5], Claude Kwe Yinda[1,5], Irene Offei Owusu[1], Myndi Holbrook [1], Robert Fischer[1],
Trenton Bushmaker [1,2], Victoria A. Avanzato[1], Jonathan E. Schulz[1], Craig Martens[3], Neeltje van Doremalen [1],
Chad S. Clancy [4] & Vincent J. Munster [1✉]

Transmission of SARS-CoV-2 is driven by contact, fomite, and airborne transmission. The relative contribution of different transmission routes remains subject to debate. Here, we show Syrian hamsters are susceptible to SARS-CoV-2 infection through intranasal, aerosol and fomite exposure. Different routes of exposure present with distinct disease manifestations. Intranasal and aerosol inoculation causes severe respiratory pathology, higher virus loads and increased weight loss. In contrast, fomite exposure leads to milder disease manifestation characterized by an anti-inflammatory immune state and delayed shedding pattern. Whereas the overall magnitude of respiratory virus shedding is not linked to disease severity, the onset of shedding is. Early shedding is linked to an increase in disease severity. Airborne transmission is more efficient than fomite transmission and dependent on the direction of the airflow. Carefully characterized SARS-CoV-2 transmission models will be crucial to assess potential changes in transmission and pathogenic potential in the light of the ongoing SARS-CoV-2 evolution.

[1] Laboratory of Virology, Division of Intramural Research, National Institutes of Health, Hamilton, MT, USA. [2] Montana State University, Bozeman, MT, USA. [3] Rocky Mountain Genomics Core Facility, Division of Intramural Research, National Institutes of Health, Hamilton, MT, USA. [4] Rocky Mountain Veterinary Branch, Division of Intramural Research, National Institutes of Health, Hamilton, MT, USA. [5] These authors contributed equally: Julia R. Port, Claude Kwe Yinda. ✉email: vincent.munster@nih.gov

Since the emergence of severe acute respiratory syndrome coronavirus-2 (SARS-CoV-2) in Wuhan, China, in December 2019, the virus has spread across the globe and has caused over 70 million cases and 1.5 million deaths as of December 2020[1]. Infection with SARS-CoV-2 can cause asymptomatic to severe lower respiratory tract infections in humans[2,3]. Peak respiratory shedding in humans occurs at the time of symptom onset or in the week thereafter. This is followed by a steady decline after the induction of a humoral immune response[4]. To a lesser extent, shedding from the intestinal tract has also been observed, but generally does not appear to be associated with the presence of infectious SARS-CoV-2 nor subsequent transmission. There is no established relationship between COVID-19 disease severity and duration and magnitude of SARS-CoV-2 shedding[5].

Considering the scale of the COVID-19 pandemic, it remains unclear to what extent the different routes of exposure contribute to human-to-human transmission and how the exposure route affects disease manifestation. In order to evaluate existing SARS-CoV-2 control measures it is crucial to understand the relative contribution of different transmission routes. Because the majority of cases have been observed in households or after social gatherings, transmission of SARS-CoV-2 is believed to be driven mostly by direct contact, fomites, and short-distance airborne transmission[6]. Airborne transmission can be defined as human-to-human transmission through exposure to large droplets and small droplet nuclei that can be transmitted through the air; whereas airborne transmission includes transmission through both large and small droplets, true aerosol transmission occurs via droplet nuclei particles smaller than 5 μm. Fomites are a result of infectious respiratory secretions or droplets being expelled and contaminating surfaces.

In multiple hospital settings SARS-CoV-2 viral RNA has been consistently detected on surfaces[7–12] and air-samples[8,9,13–20]. Detection of infectious virus in air and surface samples has been relatively limited, however infectious SARS-CoV-2 has been recovered from air samples[21] and surfaces[22,23]. Experimental research has shown viral RNA can consistently be detected for up to seven days on surfaces but, the infectious virus degrades rapidly, with limited presence after two days[12]. This discrepancy between the consistent detection of SARS-CoV-2 viral RNA and the relatively short time frames when viable virus can be detected directly hampers our ability to translate SARS-CoV-2 RNA detection on hospital surfaces and in air samples to understanding transmission and relative contribution of fomites and airborne virus.

In this work we use the well-established Syrian hamster model[24–26] to experimentally delineate the relative contribution of fomite and airborne transmission and study the impact of transmission route on disease severity using this model. We find, that aerosol inoculation causes severe respiratory pathology, higher virus loads and increased weight loss while fomite exposure leads to milder disease manifestation. Using this data, we develop a hamster airborne and fomite transmission model to confirm our findings in a natural transmission setting. Airborne transmission is more efficient than fomite transmission and dependent on the direction of the airflow. This suggests that airborne transmission may be of increased relevance in the spread of SARS-CoV-2 and highlights the relevance of targeted countermeasures.

## Results

### Clinical disease severity is correlated with exposure route. To investigate the impact of exposure route on disease severity, we compared three different inoculation routes. Three groups of 12,

4-6-week-old, female hamsters were inoculated with SARS-CoV-2 via the intranasal (I.N.; $8 \times 10^4$ TCID$_{50}$), aerosol ($1.5 \times 10^3$ TCID$_{50}$) or fomite ($8 \times 10^4$ TCID$_{50}$) routes (Fig. 1a). An unexposed control was included (N = 12) as comparison. For each group, 4 animals were euthanized on 1 day post inoculation (DPI) and 4 DPI, the remaining 4 animals were monitored until 14 DPI. Animals inoculated via the I.N. or aerosol routes demonstrated significant weight loss, whereas fomite exposure resulted in limited, transient weight loss. Animals inoculated I.N. started losing weight at 3 DPI and aerosol exposed animals at 2 DPI (Fig. 1b). Weight loss at 6 DPI was significant compared to unexposed controls for I.N., and at 4 DPI for aerosol group (Fig. 1b; N = 4, Mann–Whitney test, p = 0.0286 and p = 0.0286). In addition to weight loss, inconsistent, temporary, mild lethargy and ruffled fur were observed. Fomite exposure presented with less weight gain compared to unexposed controls. At 14 DPI no significant difference was observed between the groups (Fig. 1c; N = 4, Kruskal–Wallis test, followed by Dunn's multiple comparison test, p = 0.2953).

**Aerosol exposure directly deposits virus in the upper and lower respiratory tract, with replication in the nasal cavity epithelium, tracheal and bronchial epithelia.** Early (1 DPI) SARS-CoV-2 tropism and replication were investigated for each exposure route. Infectious virus could be detected in the trachea of all I.N. and aerosol exposed animals. In the lung, infectious virus was detected in all aerosol exposed animals and a subset of I.N. inoculated hamsters (Fig. 1d). No infectious virus was detected at 1 DPI in the fomite group in either the upper or lower respiratory tract. Compared to I.N. exposed animals, aerosol exposed hamsters demonstrated significantly increased viral load in the trachea and the lung at this time point (N = 4, ordinary two-way ANOVA, followed by Tukey's multiple comparisons test, p = 0.0115 and p = <0.0001, respectively). This suggests that aerosol exposure more efficiently deposits viral droplet nuclei in the lower respiratory system. No infectious virus was detected in the gastrointestinal tract regardless of the route of inoculation.

To investigate initial cellular tropism, immunohistochemistry (IHC) targeting the SARS-CoV-2 nucleoprotein as a marker of SARS-CoV-2 replication was performed on skull sagittal sections and lung sections at 1 DPI. In aerosol inoculated animals, viral antigen was observed in moderate to numerous ciliated epithelial cells in the nasal cavity, tracheal mucosa, and bronchiolar mucosa. In addition, viral antigen was detected in type I and type II pneumocytes, pulmonary macrophages and olfactory epithelial cells (Fig. 2a, e, i, m). Comparatively, evaluation of I.N. exposed hamsters revealed a lack of viral antigen in the epithelial cells of the trachea and lung at this timepoint. Interestingly, viral antigen was detected in pulmonary macrophages in a subset (N = 2/4) of I.N. inoculated hamsters at 1 DPI (Supplementary Fig. 1b). Viral antigen was detected in ciliated and olfactory epithelium of the nasal turbinates (Fig. 2b, f, j, m). In accordance with the virological findings, no SARS-CoV-2 antigen was detected in the trachea or lung of any fomite inoculated hamsters (N = 0/4). Viral antigen was detected in ciliated epithelial cells of the nasal turbinates in one (N = 1/4) fomite inoculated hamster (Fig. 2c, g, k, m). No SARS-CoV-2 antigen was detected in unexposed control tissues (Fig. 2d, h, l, m).

**Fomite SARS-CoV-2 exposure displays delayed replication kinetics in the respiratory tract and leads to less severe lung pathology.** To determine the correlation between exposure route and subsequent respiratory tract pathology, sections of lung, trachea and nasal turbinates were obtained for histopathological evaluation at 1 and 4 DPI. Interestingly, nasal turbinate pathology

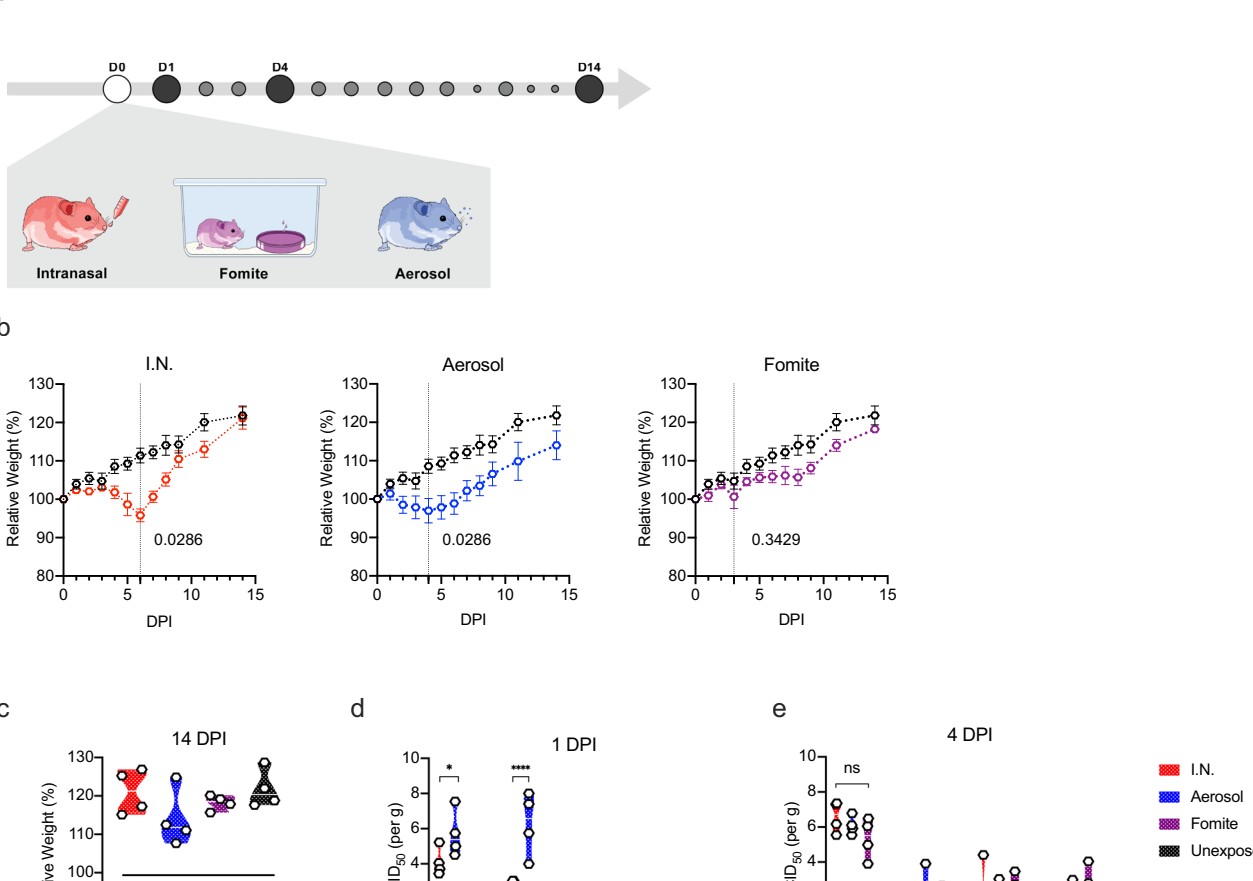

**Fig. 1 Disease severity in Syrian hamsters. a** Experimental layout for intranasal (I.N.), fomite and aerosol exposure experiments. White circle: inoculation, black: necropsy, gray, swab time-points **b** Relative weight loss in hamsters after SARS-CoV-2 inoculation over time (DPI = day post inoculation, $N = 4$ per group). The lines represent mean ± SEM. Black line indicates weights of unexposed control group. Dotted vertical line represent averaged peak weight loss post inoculation or exposure. Statistical significance was measured using a Mann–Whitney two-sided test, p-values are shown. **c** Violin plot with individuals and median of weight gain at 14 DPI. Statistical significance was measured using a Kruskal–Wallis test, followed by Dunn's multiple comparison test. **d** Violin plot with individual and median titers of infectious SARS-CoV-2 in the respiratory and intestinal tissues at 1 DPI and **e** 4 DPI, Red: I.N, blue: aerosol, purple: fomite, black: unexposed; dotted horizontal line = limit of detection (0.5). GI = gastrointestinal tract; $N = 4$ per group. Statistical significance was measured using a two-way ANOVA, followed by Tukey's multiple comparison test. *$P < 0.05$, **$P < 0.001$, ***$P < 0.0001$, ****$P < 0.0001$. NS, not significant. Source data are provided as a Source Data file.

was observed in a subset of hamsters regardless of inoculation route at 1 DPI (Fig. 3a–c). Histopathologic lesions were observed primarily in ciliated epithelial cells at 1 DPI and were most consistently observed in the I.N. inoculation group with all ($N = 4/4$) inoculated animals exhibiting mild to moderate ciliated epithelial cell necrosis with influx of numerous degenerate and non-degenerate leukocytes followed closely by aerosol inoculated hamsters with 75% ($N = 3/4$) exhibiting minimal to moderate pathology. The fomite inoculation group had the least consistent and least severe histopathologic lesions in the nasal turbinates with half ($N = 2/4$) of hamsters having no histopathologic lesions and the remaining hamsters ($N = 2/4$) having only minimal pathology. Mild to moderate tracheal inflammation was observed in all ($N = 4/4$) aerosol inoculated and half ($N = 1/2$) of the I.N. inoculated hamsters (Fig. 3e, f). Tracheal inflammation was not observed in any of the fomite inoculated hamsters ($N = 4$; Fig. 3g), confirming that virus titers detected at 1 DPI are linked

to early-onset pathological changes in this model. As expected, pulmonary pathology was minimal (aerosol and fomite) at 1 DPI, regardless of route of inoculation. Early histopathologic lesions in the lung included rare single cell bronchiolar epithelial cell necrosis, infiltration of rare or low numbers of neutrophils into the bronchiolar mucosa and focal interstitial pneumonia with minimal septal expansion by edema fluid and spillover of rare leukocytes into the adjacent alveolar spaces (Fig. 3i, j, k).

By 4 DPI, infectious virus could be detected in the lung of all animals regardless of inoculation route. No significant difference was observed between I.N. and aerosol or fomite exposed animals (Fig. 1d; $N = 4$, ordinary two-way ANOVA, followed by Tukey's multiple comparisons test, $p = 0.4114$ and $p = 0.9201$, respectively). An increase in the severity of both turbinate and pulmonary pathology was observed in all evaluated hamsters regardless of the route of inoculation. Interestingly, in both aerosol and I.N. inoculation routes, regions of olfactory epithelium within the nasal

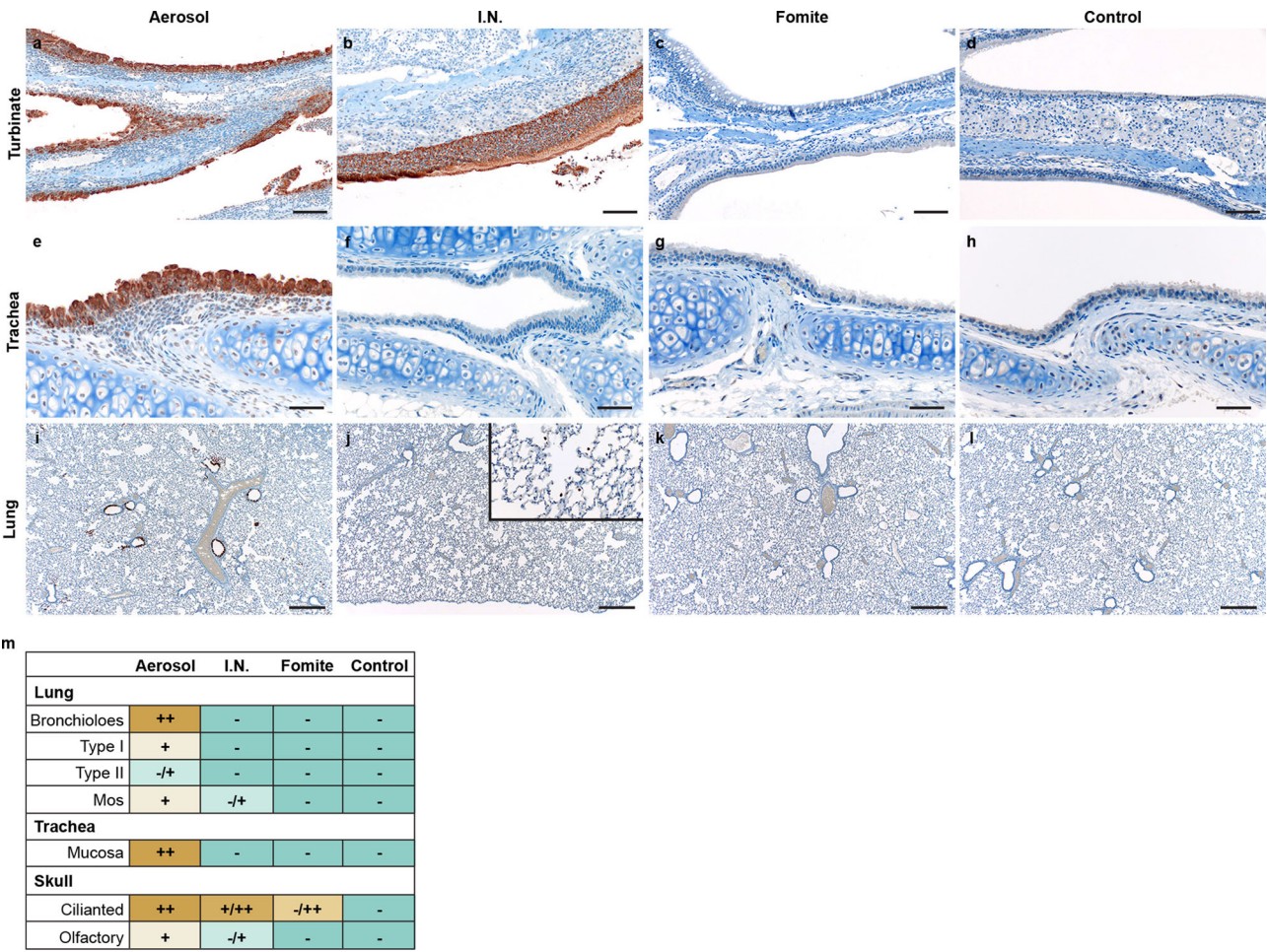

**Fig. 2 Comparison of early replication of SARS-CoV-2 in respiratory tract.** Comparison of replication of SARS-CoV-2 for intranasal (I.N.), aerosol and fomite inoculated hamsters at 1 day post inoculation (DPI) by immunohistochemistry (N = 4). **a–c** SARS-CoV-2 antigen detection in ciliated epithelial cells of the nasal mucosa (200x; bar = 100 μm). **d** Nasal mucosa from a control hamster (200x; bar = 100 μm). **e–g** SARS-CoV-2 antigen detection throughout tracheal ciliated epithelial cells (400x; bar = 50 μm). **h** Normal tracheal mucosa from a control hamster. **i** SARS-CoV-2 antigen detection focused on terminal bronchioles and adjacent alveolar spaces (40x; bar = 500 μm). **j** Lack of SARS-CoV-2 in epithelial cells with strong antigen detection noted in pulmonary macrophages (inset) (40x; bar = 500 μm). **k** Lack of SARS-CoV-2 antigen detection throughout the lung (40x; bar = 500 μm). **l** Normal lung from control hamster (40x; bar = 500 μm). **m** Quantitative comparison of antigen detection for lung (type I and type II pneumocytes, macrophages (mos), mucosa of the trachea and skull sections (olfactory and ciliated epithelium of the nasal turbinates) at 1 day post inoculation for I.N., aerosol, fomite, and control groups.

turbinates were more severely affected, suggesting initial viral attachment and replication in ciliated epithelium followed by targeting of the more caudal olfactory epithelium during disease progression (Fig. 3m–o). At this timepoint, nasal mucosal pathology was observed in all fomite inoculated animals. However, the pathology was less severe as compared to I.N. and aerosol groups and focused primarily on regions of ciliated mucosa, suggesting a delay in disease progression relative to aerosol and I.N. routes. Tracheal inflammation was observed in all inoculation routes and varied from minimal to mild (Fig. 3q–s). Moderate pulmonary pathology consistent with previously described SARS-CoV-2 infection in Syrian hamsters[24] was observed in aerosol and I.N. inoculated animals at 4 DPI (Fig. 3u, v) with less severe and less consistent pathology observed in the fomite inoculation group (Fig. 3w). Lesions were characterized as moderate, broncho-interstitial pneumonia centered on terminal bronchioles and extending into adjacent alveoli. The interstitial pneumonia was characterized by thickening of alveolar septa by edema fluid, fibrin and moderate numbers of macrophages and fewer neutrophils. Inconsistent pulmonary pathology was observed for this group with lesions ranging from minimal to moderate, which is in accordance with the observation that some fomite exposed animals did demonstrate high viral loads in the lung at 4 DPI \(Fig. 3w). No significant histopathologic lesions were observed in any control animal on 1 and 4 DPI (Fig. 3d, h, I, p, t, x).

Using a hierarchical clustering of lung pathology parameters (bronchiolitis, interstitial pneumonia, tracheitis, pathology of the ciliated and olfactory epithelium) on both 1 and 4 DPI in relation to the observed viral titers, a clear relationship existed between the respiratory pathology at 1 DPI in the trachea, and viral load of trachea and lung, while pathology in the nasal epithelial was more distantly related (Fig. 3y). Of note, viral load in the lungs at 4 DPI was most closely associated with presentation of interstitial pneumonia. Fomite exposed animals most closely resembled unexposed controls at 1 DPI and clustered together as a separate group at 4 DPI due to the appearance of tracheitis, pathology in the ciliated epithelium without distinct lower respiratory tract involvement (Fig. 3z). This implies that fomite SARS-CoV-2 exposure displays delayed replication kinetics in the respiratory tract and leads to less severe lung pathology at 4 DPI compared to direct deep deposition of virus into the respiratory tract (aerosol inoculation).

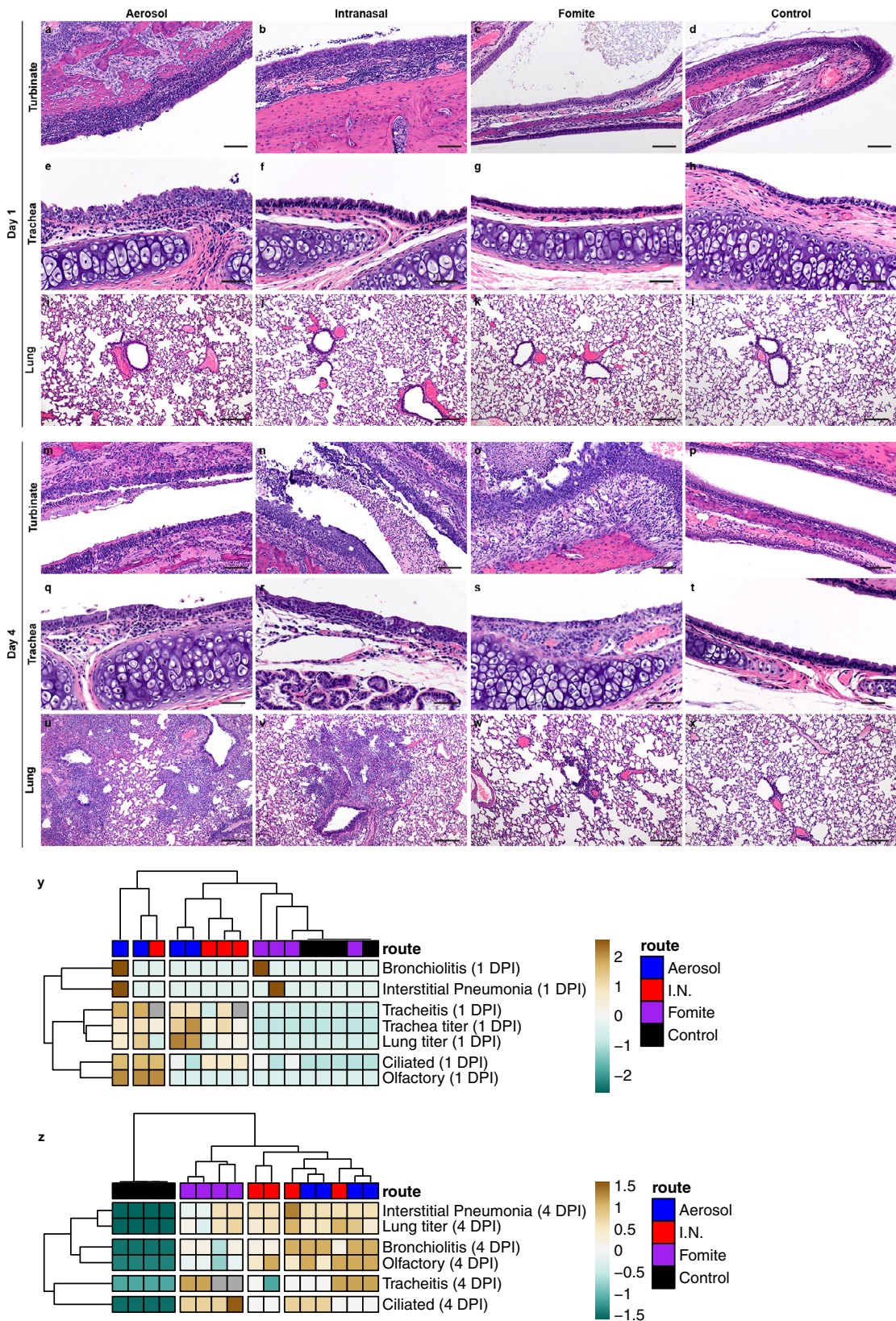

**Fomite SARS-CoV-2 exposure results in a reduced immune profile in the lung**. To investigate the systemic immune response, cytokine specific ELISAs were performed on serum at 4 DPI (Fig. 4a). While overall significance was low, serum levels were different depending on exposure route for pro-inflammatory tumor necrosis factor (TNF)-α and anti-inflammatory interleukin

(IL)−4 and IL-10. In contrast to unexposed, I.N. and aerosol groups, fomite exposed animals presented with decreased levels of TNF-α at 4 DPI; a significant difference in serum levels was detected between I.N. and fomite exposed groups ($N = 4$, Kruskal–Wallis test, followed by Dunn's multiple comparisons test, $p = 0.0360$). Adversely, the IL-4 levels were increased in all

**Fig. 3 Comparison of the respiratory tract pathology of SARS-CoV-2 Infected hamsters.** Comparison of SARS-CoV-2 pathology for intranasal (I.N.), aerosol and fomite inoculated hamsters at 4 day post inoculation (DPI) ($N = 4$). **a** Infiltration and disruption of the ciliated nasal mucosa by moderate numbers of leukocytes with multifocal epithelial cell necrosis (200x; bar = 100 μm). **b** Multifocal disruption of the nasal ciliated mucosa by low numbers of leukocytes with accumulations of degenerate leukocytes in the nasal passage (200x; bar = 100 μm). **c** Intact ciliated nasal mucosa with normal mucus presence within the lumen (200x; bar = 100 μm). **d** A control nasal turbinate with intact ciliated nasal mucosa and mucus within the lumen (200x; bar = 100 μm). **e** Disruption of the tracheal mucosa with single cell necrosis and infiltration by low numbers of leukocytes (400x; bar = 50 μm). **f** Unaffected tracheal mucosa (400x; bar = 50 μm). **g** Unaffected tracheal mucosa (400x; bar = 50 μm). **h** Section of tracheal mucosa from a control hamster (400x; bar = 50 μm). **i-l** No significant histopathologic lesions in the lung of any inoculation route at 1 day-post-inoculation (100x; bar = 200 μm). **m** Multifocal disruption of ciliated nasal mucosa with accumulation of cellular debris and degenerate leukocytes within the nasal passage (200x; bar = 100 μm). **n** Severe disruption and multifocal erosion of the nasal mucosa with accumulation of numerous degenerate leukocytes and abundant cellular debris within the nasal passage (200x; bar = 100 μm). **o** Ciliated epithelial cell degeneration and mucosal erosion with leukocyte infiltration into the lamina propria (200x; bar = 100 μm). **p** Normal nasal turbinate from a control hamster (200x; bar = 100 μm). **q** Focal disruption of the tracheal mucosa by low numbers of leukocytes (400x; bar = 50 μm). **r** Multifocal infiltration of the mucosa by moderate numbers of leukocytes and multifocal epithelial cell necrosis (400x; bar = 50 μm). **s** Multifocal loss of epithelial cilia and infiltration of the lamina propria by moderate numbers of leukocytes (400x; bar = 50 μm). **t** Normal tracheal mucosa from a control hamster (400x; bar = 50 μm). **u** Widespread, moderate to severe broncho-interstitial pneumonia (100x; bar = 200 μm). **v** Multifocal moderate broncho-interstitial pneumonia focused on terminal bronchioles (100x; bar = 200 μm). **w** Multifocal, mild interstitial pneumonia focused on terminal bronchioles (100x; bar = 200 μm). **x** Normal lung from a control hamster (100x; bar = 200 μm). **y, z** Clustering (Euclidean, complete) of animals based on viral titers in lung and trachea and quantitative assessment of pathology in the upper and lower respiratory tract on 1 DPI and 4 DPI. Heatmap colors refer to color scale on the right, gray = NA, I.N. = red, Aerosol = blue, Fomite = purple, Control = black. Exposure route is indicated by color bar at the top. Source data are provided as a Source Data file.

groups as compared to unexposed animals, yet highest levels were seen in fomite exposed animals, the difference between unexposed and fomite group reaching statistical significance ($N = 4$, Kruskal–Wallis test, followed by Dunn's multiple comparisons test, $p = 0.0109$). Increased serum IL-10 was also observed in fomite exposed animals and I.N. exposed animals, while a decrease was observed in animals after aerosol exposure, resulting in a significant difference between aerosol and fomite exposed hamsters ($N = 4$, Kruskal–Wallis test, followed by Dunn's multiple comparisons test, $p = 0.0286$). While not significant, a trend of decreased serum levels of interferon (INF)-γ as compared to uninfected animals, was observed. No significant differences were seen for serum levels of IL-6.

Irrespective of exposure route, all exposed animals seroconverted at 14 DPI as seen by the presence of antibodies targeting the SARS-CoV-2 spike measured by ELISA (Fig. 4b). The magnitude of humoral response was linked to the exposure route. I.N. exposure resulted in the strongest, and significantly higher antibody response when compared to fomite exposure ($N = 4$, Kruskal–Wallis test, followed by Dunn's multiple comparisons test, $p = 0.0209$). No significant difference was observed between I.N. and aerosol exposed animals. We compared the neutralizing capacity against live virus. Aerosol exposed animals demonstrated highest neutralizing titers and fomite exposed animals lowest, however not to significant difference (Fig. 4c, $N = 4$, Kruskal–Wallis test, followed by Dunn's multiple comparisons test, $p = 0.2026$), and the ratio between neutralizing titers and ELISA titers did equally show no difference (Supplementary Fig. 1e).

To gain insight into the local immune responses in the lung, we evaluated global changes in the gene expression at 1 and 4 DPI in comparison to control animals. Three lung samples were removed due to quality issues (Supplementary Table 1). Principal components analysis revealed expected grouping from most conditions with each group containing their associated replicates. The largest separation was from groups 1 and 4 DPI aerosol samples, followed by less separation between the remaining six conditions. However, within this second cluster of the remaining six conditions, separate ellipses representing two standard deviations can still be viewed as non-intersecting groups, distinct from the controls (Supplementary Fig. 2).

To assess which pathways were differently regulated for each exposure route, the gene expression information was imported into Integrated Pathway Analysis (IPA) software. The results show that in I.N. and aerosol exposed groups over 50 canonical pathways were up- or downregulated significantly in comparison to control animals (p-value < 0.05, z-score < −2 or > 2); amongst which metabolic, immune, infection and cell function associated pathways (Fig. 4d, Supplementary Table 2 shows all significant pathways). In fomite animals, only 10 pathways were found to be significantly up- or downregulated as compared to control animals. In I.N. and aerosol exposed animals, pathway analysis revealed macrophage activation, dendritic cell maturation, interferon signaling and T-, B- and NK-cell involvement. In fomite exposed animals the interferon signaling, Th17 pathway and pattern recognition for bacteria and viruses were upregulated. Interestingly, involvement of the Th17 pathway were found in all three.

As we saw similar virus titers in the lungs of animals at 4 DPI, using IPA we next compared specifically the virus-induced response (coronavirus pathway) and the resulting adaptive immune response (Th1/Th2 pathway) in more detail (Fig. 4e). Aerosol and I.N. exposed animals showed differential expression and regulation of genes associate with the coronavirus pathway. In comparison to I.N. exposed animals, aerosol exposure at 4 DPI was linked to downregulation of multiple key mediators including MAVS, ELK1, BCL2, Serpine 1 and IFNAR1, but upregulation of IL-1b. In comparison, we found no major differential expression of the coronavirus pathway associated genes in fomite exposed animals. In detail, the Th1/Th2 pathway upregulation comprised in I.N. and aerosol animals amongst others increased gene expression of pro-inflammatory cytokines IL-18, IFN-γ, IL-6 and IL-2, upregulation of expression of surface molecules CD4 and CD8, multiple co-activation molecules like CD28, CD80, CD40 and chemokine receptors and tissue trafficking receptors such as CXCR3, CXCR6, CCR8 and ITGB2. In contrast, fomite exposed animals showed minimal upregulation and clustered closest to controls. Taken together this suggests a predominantly mild immune response, as compared to aerosol exposure, is mounted after fomite exposure in the lung which may protect from more severe outcome.

**Viral shedding is exposure route dependent**. To gain an understanding of route-dependent virus shedding patterns of SARS-CoV-2 in the Syrian hamster, daily oropharyngeal and

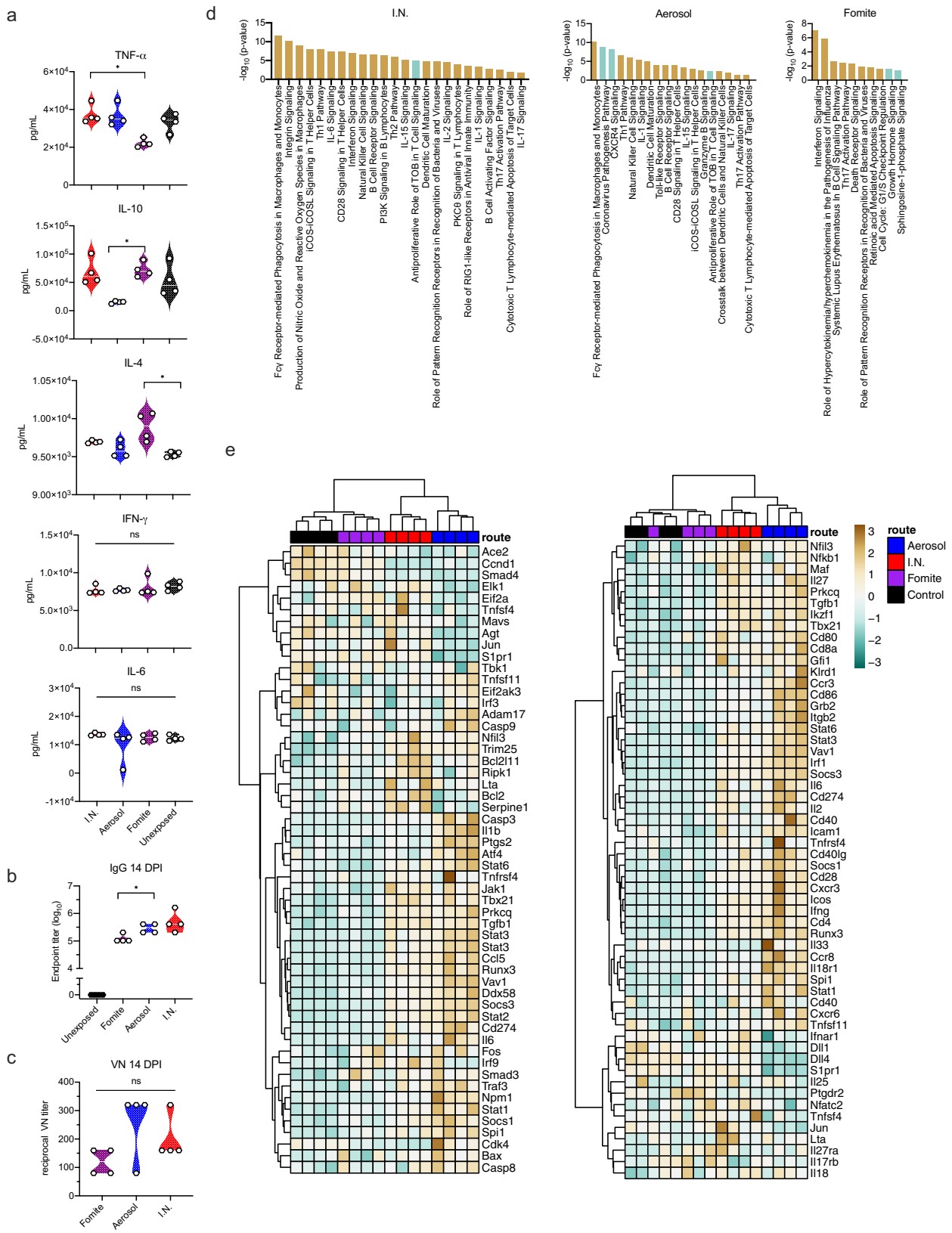

rectal swabs were taken until 7 DPI, after which swabs were taken thrice weekly (Fig. 5a, b, Supplementary Fig. 1f). Oropharyngeal swabs are a measurement of respiratory shedding while rectal swabs assess intestinal shedding. Viral sgRNA, a marker of virus replication[27], was detected in both swabs from all exposed animals on at least one day. When comparing the overall respiratory

shedding profile between the exposure routes, different patterns were observed. I.N. inoculation resulted in high viral loads starting at 1 DPI and continued up until 6 DPI, before sgRNA levels started to decrease. In the aerosol inoculated group, the peak of virus shedding was reached on 2 DPI and viral sgRNA levels decreased immediately thereafter. In contrast, animals

**Fig. 4 Exposure dependent SARS-CoV-2 acute local immune gene activation, systemic cytokine response and strength of humoral response. a** Violin plots with individuals and median of serum concentrations of key cytokines (interferon (IFN)-γ, tumor necrosis factor (TNF)-α, interleukin (IL)-6, IL-4, and IL-10) on 4 days post inoculation (DPI). Statistical significance was measured using a Kruskal–Wallis test. **b, c** Violin plots with individuals and median of endpoint IgG antibody titers against SARS-CoV-2 spike ectodomain measured by ELISA in serum and reciprocal live virus neutralization titers. ELISAs and neutralization assays (VNs) were done once. **d** Selection of significantly up- (brown) or downregulated (blue) immune- or infection associated pathways in the lung at 4 DPI, identified by integrated pathway analysis. **e** Clustering (Euclidean, Ward.D2) of animals based on gene-expression associated with the coronavirus pathway (left) and Th1/Th2 pathway (right) in lung at 4 DPI. Heatmap colors refer to color scale on the right (normalized z-score). Exposure route is indicated by color bar at the top, I.N. = red, Aerosol = blue, Fomite = purple, Control = black. *P < 0.05, **P < 0.001, ***P < 0.0001, ****P < 0.0001. NS, not significant. Source data are provided as a Source Data file.

exposed through the fomite route demonstrated different shedding kinetics as compared to aerosol and I.N. groups with an increase in viral sgRNA shedding over multiple days, until peak shedding was reached at 5 DPI. While a trend seemed present for higher individual peak shedding in I.N. and fomite groups, no significant difference was detected (Fig. 5c; N = 4, Kruskal–Wallis test, followed by Dunn's multiple comparisons test, p = 0.8400). In comparison, intestinal shedding demonstrated median lower viral loads with no significant difference between groups: N = 4 Kruskal–Wallis test, followed by Dunn's multiple comparisons test, p = 0.1512 (Fig. 5b, d). Looking at the shedding profile of individual animals across groups, intestinal shedding was observed for a maximum of three consecutive days with sgRNA only being detected in swabs for one or two consecutive days for most positive animals. To evaluate the overall shedding burden generated by each exposure route, the cumulative shedding up until 14 DPI (area under the curve (AUC)) was compared. Aerosol exposure led to overall less viral RNA in oropharyngeal swabs as compared to I.N. and fomite exposure (N = 4, Kruskal–Wallis test, followed by Dunn's multiple comparisons test, p = 0.0263). In contrast, most cumulative viral sgRNA was detected in rectal swabs of aerosol exposed animals (Fig. 5d). Taken together, these data suggest that severity of disease is not indicative of the duration and cumulative amount of virus shed after infection.

**Early shedding profile predicts disease severity and corresponding immune response.** As we observed different impacts on disease profiles between exposure routes, we next investigated potential predictability of disease through early shedding patterns. Cytokine responses as a measurement of the immune status (4 DPI) were included in the correlations between early shedding (2 DPI), peak shedding, peak weight loss, lung titers and pathology at 4 DPI, antibody titers and neutralizing capacity at 14 DPI (Fig. 4e). Lung viral titers were positively correlated significantly with the amount of viral RNA detected in oropharyngeal swabs at 2 DPI (Spearman correlation test, N = 12, p = 0.047). Lung titers showed a positive relationship with upper and lower respiratory tract pathology and weight loss. This suggests that early time point respiratory shedding (before disease manifestation) may predict the acute disease manifestation.

Serum levels of IL-4, IL-6 and IL-10 did not show any significant correlations with parameters of disease severity; however, a clear negative relationship could be seen in the correlations. TNF-α, negatively correlated to IL-4 and IL-10 levels (Spearman correlation test, N = 12, p = 0.048 and p = 0.008, respectively). A positive correlation between early rectal shedding and TNF-α serum levels and olfactory pathology was observed (Spearman correlation test, N = 12, p = 0.0002 and p = 0.001, respectively). Interestingly, olfactory pathology also showed positive correlation to the magnitude of the IgG response and the neutralizing capacity (Spearman correlation test, N = 12, p = 0.001 and p = 0.021, respectively) (Fig. 5e).

**Airborne transmission is more efficient than fomite transmission in the Syrian hamster.** To investigate viral fomite contamination of caging, daily swabs were taken from surfaces in cages containing one I.N. inoculated hamsters, up to 7 DPI (Supplementary Fig. 1c, d). Viral gRNA was detectable at 1 DPI in all samples, sgRNA was detectable for 7/8 (87.5%) bedding samples and 3/8 (37.5%) cage samples, and at 2 DPI in 8/8 cages for both samples. Viral sgRNA was detectable at high concentrations up until 7 DPI, with peak concentrations seen on 2 and 3 DPI, suggesting a robustly contaminated caging environment.

To assess the potential risk of fomite transmission, we introduced sentinel hamsters to cages after housing two I.N. infected animals for 4 days. (Fig. 6a). No signs of disease or weight loss were observed in sentinel animals, but seroconversion was seen in 4 out of 8 animals (Fig. 6f) at 21 days after exposure (DPE) to a contaminated cage, confirming that hamster-to-hamster indirect transmission via fomites can occur (Fig. 6h).

Next, the efficiency and dynamics of airborne hamster-to-hamster transmission were assessed. For this purpose, we designed a cage divider, which allowed airflow but no direct contact or fomite transmission between animals. (Fig. 6b, c, d, and Supplementary Movie 1). We used a particle sizer to assess the effect of the cage divider on blocking particle flow. We observed that cross-over of smaller particles (<10 μm) was blocked approx. 60%, whilst larger particles (>10 μm), were reduced over 85% on the sentinel side (Fig. 6d, e).

In the first experiment, one sentinel hamster was placed on the side of the divider downflow from one infected animal (N = 8). In contrast to animals exposed directly to aerosolized virus, no signs of disease or weight loss were observed in any of the sentinel animals (Fig. 6g). However, all animals seroconverted. To assess the importance of directional airflow, airborne transmission was also modeled for 4 transmission pairs housing the sentinel against the airflow (Fig. 6b, c). Only one out of 4 of the sentinels placed against airflow seroconverted (Fig. 6h), suggesting, as expected, that directional airflow is key to airborne transmission. When comparing the antibody response at 21 DPI/DPE, no significant difference was determined between the donor I.N. inoculated animals and those that seroconverted after airborne transmission (100%), while titers for animals that seroconverted after fomite transmission (50%) were lower (Fig. 6h, Kruskal–Wallis test, followed by Dunn's multiple comparisons test, N = 8 and N = 4, p = >0.9999 and p = 0.2488, respectively). Titers were comparable to those observed after direct I.N. inoculation. Together, this suggests that hamster-to-hamster airborne transmission may present with asymptomatic disease manifestation, yet the humoral immune memory is comparably robust.

To investigate the transmission risk posed by animals after fomite or airborne transmission, the respiratory shedding profile was determined. Viral shedding was demonstrated in 4 out of 8 sentinel hamsters after exposure to contaminated cages on multiple consecutive days. Shedding was observed at 1 DPE, with peak viral sgRNA being seen at 4/5 DPE, like what was observed

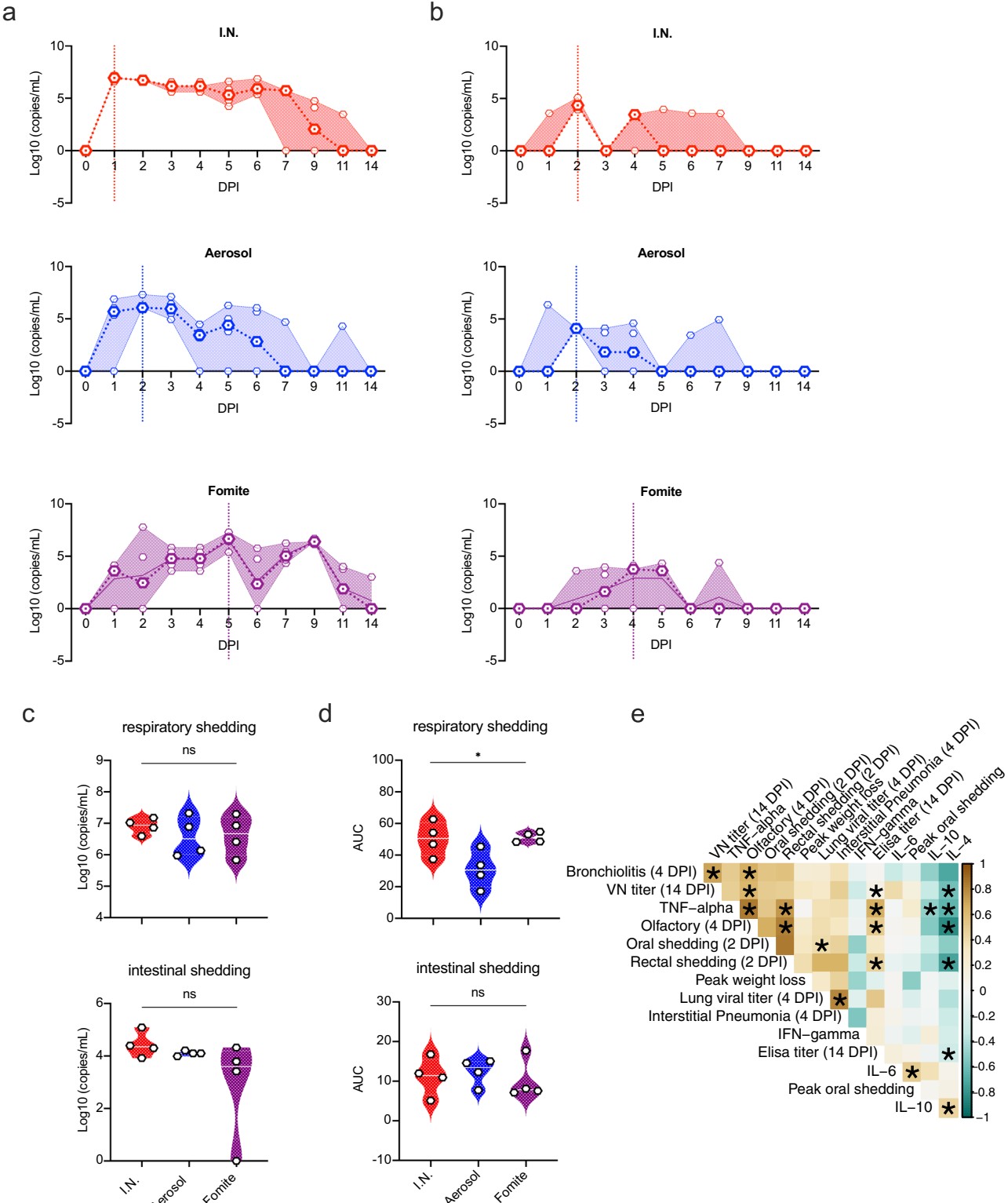

**Fig. 5 Exposure dependent SARS-CoV-2 shedding. a** Respiratory and **b** intestinal viral shedding of intranasal (I.N.), aerosol and fomite exposed hamsters. Median, 95% CI and individuals are shown. **c** Peak shedding and **d** cumulative (area under the curve (AUC) analysis) respiratory and intestinal shedding of I.N., aerosol and fomite exposed hamsters. Statistical significance was measured by Kruskal–Wallis test, $N = 4$ per group. $*P < 0.05$, $**P < 0.001$, $***P < 0.0001$, $****P < 0.0001$. NS, not significant. **e** Correlation between cytokine levels, early shedding (2 days post inoculation (DPI)), peak shedding, peak weight loss, ELISA and virus neutralization (VN) titers (14 DPI), lung titers and pathology at 4 DPI. Significant correlations ($N = 4$ per group, Pearson–Spearman analysis, $p < 0.05$) are indicated with an asterisk and strength of correlation ($R^2$) is depicted according to the color bar on the right. Source data are provided as a Source Data file.

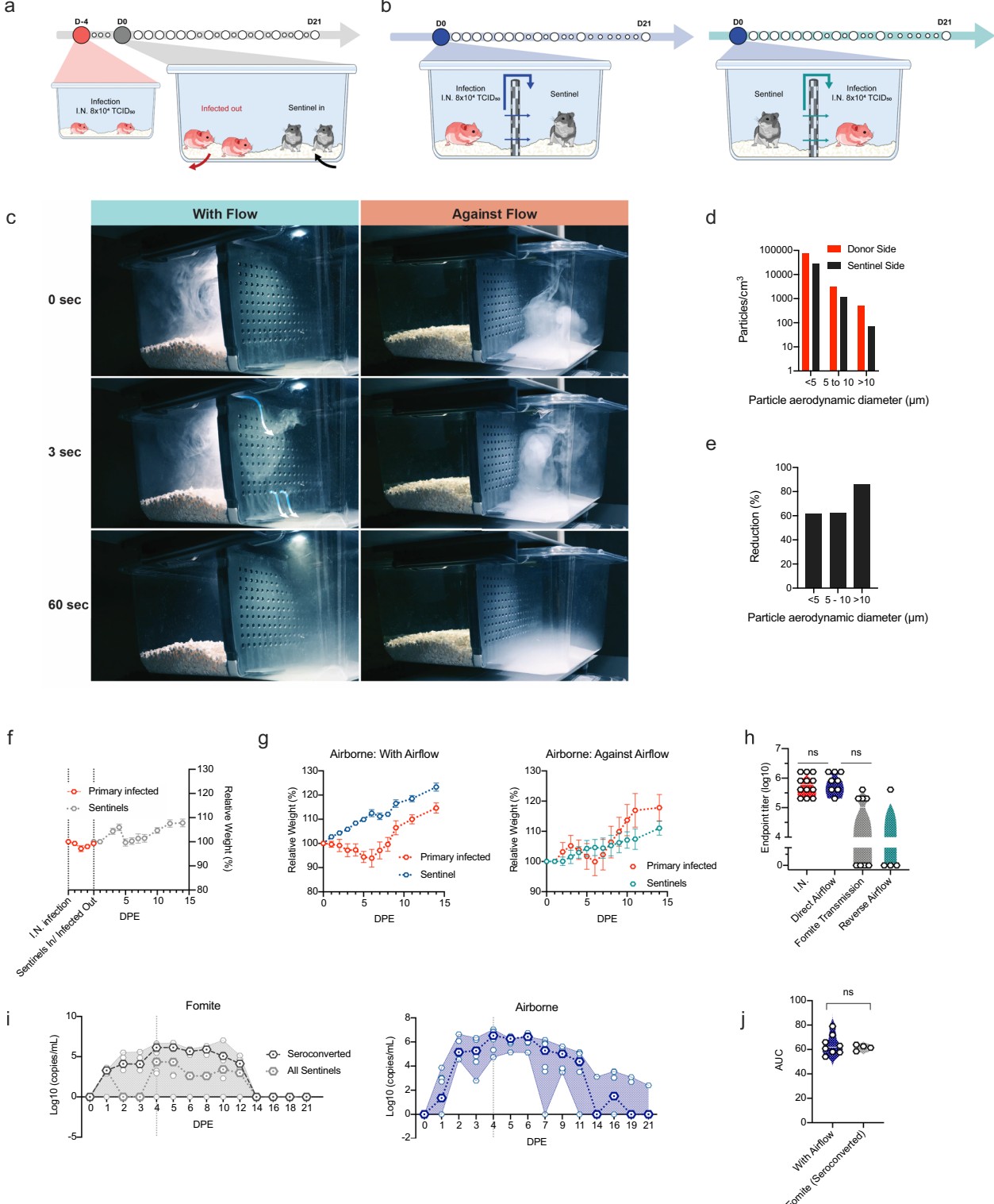

**Fig. 6 Fomite and airborne transmission in the Syrian hamster. a** Experimental layout for fomite and **b** airborne exposure experiments in hamsters. I.N. = intranasal. **c** Pictures of smoke test to demonstrate unidirectional airflow in the transmission cage. **d** Aerodynamic particle size distribution on either side of the transmission cage. **e** Reduction of particles by the divider. **f, g** Relative weight loss in hamsters after SARS-CoV-2 transmission via fomite and airborne routes. DPE = days post exposure. Lines represent mean ± SEM. **h** Violin plot with individuals and median of endpoint IgG antibody titers against SARS-CoV-2 spike ectodomain by ELISA in serum of hamsters infected through airborne and fomite transmission route. ELISAs were done once. **i** Respiratory shedding profile of hamsters exposed through fomite and airborne transmission routes, individuals, median and 95% CI are shown. **j** Cumulative (area under the curve (AUC) analysis) of respiratory shedding from animals which seroconverted after airborne and fomite transmission. Violin plots with individuals and median are depicted. Statistical significance was measured by Kruskal–Wallis test, $N = 8$ per group. $*P < 0.05$, $**P < 0.001$, $***P < 0.0001$, $****P < 0.0001$. NS, not significant. Source data are provided as a Source Data file.

in hamsters directly exposed to fomites (Fig. 5a). For airborne transmission, sentinels downstream of airflow started shedding by 1 DPE, and all 8 animals had high amounts of viral sgRNA in the oropharyngeal cavity by 2 DPE, which remained high until 6 DPE. This data suggest that this indirect exposure route presents with a distinctly different disease manifestation and shedding profile than direct aerosol exposure (Fig. 6i). Of note, commutative viral shedding between infected airborne exposed animals showed no difference to those infected through fomite transmission (Fig. 6j). These data imply that, whilst presenting with no or very mild disease phenotypes, both routes of indirect exposure between animals create a mimicry of asymptomatic carriers.

## Discussion

SARS-CoV-2 transmission is driven by close proximity, confined environment, and the frequency of contacts[28]. Infection with SARS-CoV-2 is believed to be driven by direct contact, inhalation of virus within respiratory droplet nuclei, contact with droplet contaminated surfaces or any combination between these exposures. Yet, the relative contribution of each of the potential routes of exposure in relationship to human-to-human transmission has been elusive. Moreover, the relationship between exposure route and dose and the differential impact on disease severity has been equally obscure. Animal models are essential to model experimental transmission under controlled conditions, as transmission involves several factors: duration and magnitude of virus shedding, stability of the virus in aerosols or on surfaces, and the subsequent infection of another host.

Our data suggest that in addition to the exposure dose[29] and underlying host conditions[30], disease is also a function of exposure route. The Syrian hamster model recapitulates several aspects of COVID-19, including upper and lower respiratory tract pathology, SARS-CoV-2 shedding and potential transmission between animals[24–26,29,31–33]. Typically, experimental studies with SARS-CoV-2 in hamsters rely on intranasal inoculation. This route of inoculation establishes robust infection but does not resemble natural infection via aerosols or respiratory droplets. Here we directly compared natural transmission routes, designed to mimic airborne and fomite exposure by presenting the first data on aerosol and direct fomite inoculation in this model.

The initial respiratory tropism of SARS-CoV-2 was determined by the exposure route, aerosol exposure deposited SARS-CoV-2 more efficiently in both the upper and lower respiratory tract. As a result, the SARS-CoV-2 replication kinetics displayed higher viral titers in trachea and lung early in the infection compared to the I.N. inoculated animals. While SARS-CoV-2 RNA was detected by PCR of tracheal tissue at 1 DPI from I.N. inoculated animals, SARS-CoV-2 N protein antigen was not detected by IHC at this timepoint. Several factors may contribute to this: proximal trachea was taken for molecular and virologic analysis, which is the site in closest physical proximity to the site of inoculation. This site may have been infected more rapidly due to the proximity to the inoculation dose. Additionally, IHC can be a highly specific assay but may be poorly sensitive during the pre-acute phase of disease when little viral antigen is being produced. Despite a 10-fold lower inoculation dose, exposing Syrian hamsters to aerosolized SARS-CoV-2 resulted in more rapid virus replication in the lung and weight loss compared to I.N. inoculation. In contrast, fomite inoculation displayed a prolonged time between exposure and viral replication in the lung leading to reduced disease severity. This delay suggests that for fomite infection viral replication may occur in the oropharynx before being inhaled[32]. It is therefore possible that initial immune priming occurs not in the lower respiratory tract and that this may give time for a regulating immune response characterized by

a systemic lack of TNF-α. The decrease in TNF-α may reduce immune pathology in the lung even with the observed viral titers at 4 DPI not being significantly lower as compared to aerosol inoculation. It has been previously shown that IL-10 and IL-4 may be regulated in a differentiated and infection-route dependent manner, which would explain the differential systemic presence of these cytokines here[34]. However, our systemic analysis of cytokine profiles was severely limited by the availability of hamster-specific reagents and remains superficial. Our data on the local transcriptome at 4 DPI demonstrates that aerosol and I.N. exposure led to major and differential involvement of coronavirus pathway associated genes and increased the upregulation of a subset of genes in the Th1 and Th2 response pathways, whereas fomite exposure resulted in minimal pathway activation at this time point. This could further explain, why in these animals we observed increased pathology, which is absent in the fomite animals. Previously, single-cell analysis in the hamster has demonstrated that inflammation and CD4+ and CD8+ cytotoxic T-cell responses preceded viral elimination in the hamster[35]. The gene expression profile seen after aerosol and I.N. exposure would also suggest increased T-cell, NK cell and macrophage recruitment to the site of infection, as well as activation of the humoral response. Interestingly, fomite exposed animals still mounted a considerable humoral response, which could further imply that the immune response in these animals may be driven by infection outside of the lung. If these differential immune-signatures are not only a result of dose-kinetic but an intrinsic effect of the exposure routes, this could have important implications in the context of re-infection with novel variants. We acknowledge that these direct exposure experiments were performed using cultured virus and that it cannot be ruled out that the particle to infectivity ratio may differ from those found in naturally shed samples. Additionally, while dosage of I.N. and aerosol exposure could be accurately determined, the caveat must be given, that the exact dosage through fomite could not be confirmed. While previous work in this model has demonstrated that disease severity and shedding profile is not overtly affected by infection dose after I.N. exposure[24], we cannot confirm that the dose-dependency after aerosol and fomite exposure may not present differently.

No human data is currently available on the influence of transmission route on COVID-19 severity. In experimental Nipah virus infection studies in non-human primates, particle size directly influenced the disease manifestations. Aerosol exposure led to a rapidly progressing respiratory disease whereas large droplet exposure led to an extended disease course that does not have the prominent respiratory features[36,37]. These findings suggest more severe disease is associated with direct deposition of the virus in the lower respiratory tract, whereas with milder disease the first viral replication occurs in upper respiratory tract. This further implies that besides lowering viral dose, intervention measures such as face-coverings may also serve to minimize disease by limiting the deposition of viral particles into the lower respiratory tract[38–40]. More investigations are required to validate if this occurs[41].

Our data reflects findings in humans, where no clear correlation could be drawn between severity of disease and shedding time. The aerosol exposed animals shed cumulatively less virus, while fomite exposure resulted in equally high peak viral shedding compared to I.N. inoculated animals. In humans, serological analyses suggest that approximately 17% of infections remain mild to asymptomatic[42]. There is evidence of both asymptomatic and symptomatic shedding[43–46], suggesting that mild or asymptomatic disease contribute the same transmission risk as more severe COVID-19 cases[47,48]. Asymptomatic disease in humans may present with lower shedding dose or faster decline[5], which we did not observe in this animal model.

The relative contribution of fomite and airborne transmission to the spread of SARS-CoV-2 is still disputed[49]. The risk of fomite transmission was previously assessed as lower compared to airborne transmission in a limited study in the Syrian hamster. Fomite transmission occurred in only 1 of 3 sentinels placed into contaminated cages at viral RNA peak contamination[25]. Surprisingly, we demonstrate here that fomite transmission may still occur (4 out of 8) when peak shedding of infectious virus has waned as previously shown[25], and environmental contamination is expected to be reduced. Importantly, this implies that even with an increased understanding of airborne transmission involvement at this stage of the pandemic, the risk of fomite transmission in humans should not be underestimate. There has been considerable effort to culture virus samples from contaminated surfaces or air, yet there is significant discrepancy between viral RNA detected and actual isolation of live virus[8,14,50]. This may be due to a lack of efficient culture methods or a result of overestimation of the viral environmental burden based on PCR methodology. As we assessed the environmental contamination in hamster cages by PCR and found it consistently high even after shedding of infectious virus by the donors is expected to have ceased[25], this may suggest a transmission risk even when culturable virus may not be found. Additionally, it needs to be acknowledged, that the hamster and human interaction with potential fomites is not equal. While hamsters can be assumed to interact with their environment in more intimate and consistent manner, the risk of fomite exposure demonstrated in this study does highlight the important role of reducing tactile interactions with potentially contaminated surfaces also for humans. Further, it does support countermeasures such as hand-washing and regular decontamination of surfaces, which are likely to reduce the risk of infection[51]. In particular, fomite transmission may be more likely to occur in nosocomial settings that present a combination of fomite and aerosol generating procedures and may potentially be further enhanced with more susceptible hospital populations[52,53].

Within our transmission set-up we show a selective reduction of largest particles (>10 μm), but that this exclusion was not absolute (Fig. 5). Therefore, we cannot formally distinguish between true aerosol transmission (droplet nuclei < 5 μm), droplet transmission (> 10 μm), or a combination of these two. Previous studies have shown that SARS-CoV-2 can be transmitted through the air in a ferret model over short and moderate distance[54,55] and in hamsters over short distance[25,56]. In our study we were able to show a high efficiency of airborne transmission with 100% of the sentinels becoming infected. When reversing the airflow from uninfected animals toward infected animals, a sharp reduction in transmission was observed. This suggest that directional airflow plays an important role in the transmission of SARS-CoV-2. This has also been observed in human-to-human transmission events, where transmission in confined spaces (e.g. restaurant) was directed by airflow[16,57,58]. The results of the experiments with directional and reverse-directional airflow provide direct experimental data supporting preemptive SARS-CoV-2 control measures focused on improvement of ventilation[59,60].

In this study, we showed the relative contribution of airborne and fomite transmission and the impact of exposure route on disease. The hamster transmission model will be crucial to assess the transmission and pathogenic potential of novel SARS-CoV-2 strains, in the light of the continuing SARS-CoV-2 virus evolution[61]. In addition, this work will allow the development of effective public health countermeasures aimed at blocking human-to-human transmission. The findings of this study suggest that using more natural routes of transmission is highly suitable for accurately assessing the transmission potential and pathogenicity of novel evolved strains[61]. Novel variants such as B.1.1.7 and B.1.351 are currently replacing the old circulating variants. Shedding patterns in humans suggest increased transmissibility may not only be a function of increased binding capacity to hACE2 or replication[62–64]. If these variants are shown to have different environmental stability, it would be prudent to also investigate if this impacts exposure routes. Additionally, these data strongly suggest that the Syrian hamster model would be very suitable to investigate the true limits of airborne transmission and applying this to prevention studies as has been previously demonstrated for short distance airborne transmission with masks[56]. Furthermore, demonstrating hamster-to-hamster natural transmission via different routes indicates that this model is useful for setting up complex intervention experiments involving different transmission routes.

## Methods

**Ethics statement**. Approval of animal experiments was obtained from the Institutional Animal Care and Use Committee of the Rocky Mountain Laboratories. Performance of experiments was done following the guidelines and basic principles in the United States Public Health Service Policy on Humane Care and Use of Laboratory Animals and the Guide for the Care and Use of Laboratory Animals. Work with infectious SARS-CoV-2 strains under BSL3 conditions was approved by the Institutional Biosafety Committee (IBC). Inactivation and removal of samples from high containment was performed per IBC-approved standard operating procedures.

**Virus and cells**. SARS-CoV-2 strain nCoV-WA1-2020 (MN985325.1) was provided by CDC, Atlanta, USA. Virus propagation was performed in VeroE6 cells in Dulbecco's Modified Eagle Medium (DMEM) supplemented with 2% fetal bovine serum (FBS), 2 mM L-glutamine, 100 U/mL penicillin and 100 μg/mL streptomycin. Cells were cultured in DMEM supplemented with 10% FBS, 2 mM L-glutamine, 100 U/mL penicillin and 100 μg/mL streptomycin. No contaminants were detected; the used virus was 100% identical to the initial deposited GenBank sequence (MN985325.1).

**Inoculation experiments**. Four to six-week-old female Syrian hamsters (ENVIGO) were inoculated (12 animals per route) either intranasally (I.N.), via aerosol exposure or via exposure to a fomite. Hamsters were housed in groups of 4 animals. I.N. inoculation was performed with 40 μL sterile DMEM containing $8\times10^4$ TCID$_{50}$ SARS-CoV-2. For exposure through aerosols animals were subjected to $1.5\times10^3$ TCID$_{50}$ SARS-CoV-2 during a 10 min exposure time. Aerosol inoculation using the AeroMP aerosol management platform (Biaera technologies, USA) was performed as described previously[65]. Briefly, non-anesthetized hamsters were exposed to a single exposure whilst contained in a stainless-steel wire mesh cage. Aerosol droplet nuclei were generated by a 3-jet collision nebulizer (Biaera technologies, USA) and ranged from 1-5 μm in size. A sample of 6 liters of air per min was collected during the 10 min exposure on the 47 mm gelatin filter. Post exposure, the filters were dissolved in 10 mL of DMEM containing 10% FBS and infectious virus was titrated as described above and the aerosol concentration was calculated. The estimated inhaled inoculum was calculated using the respiratory minute volume rates of the animals determined using the methods of Alexander et al.[66]. Weights of the animals were averaged and the estimated inhaled dose was calculated using the simplified formula $D = R \times C_{aero} \times T_{exp}$[67], where $D$ is the inhaled dose, $R$ is the respiratory minute volume (L/min), $C_{aero}$ is the aerosol concentration (TCID$_{50}$/L), and $T_{exp}$ is duration of the exposure (min). Fomite exposure was conducted by placing a polypropylene dish into the cage containing 40 μL of $8\times10^4$ TCID$_{50}$ SARS-CoV-2 per hamster (total dose per cage: $3.2\times10^5$ TCID$_{50}$) for 24 h. Interaction of hamsters with the dish was monitored and confirmed within the first 5 minutes after placing it into the cage. For I.N. and fomite exposure undiluted stock virus with confirmed dose was applied.

At 1- and 4-days post infection (DPI), four hamsters for each route were euthanized, and tissues were collected. The remaining 4 animals for each route were euthanized at 14 DPI for disease course assessment and shedding analysis. Hamsters were weighted daily, and oropharyngeal and rectal swabs were taken daily until day 7 and then thrice a week. Swabs were collected in 1 mL DMEM with 200 U/mL penicillin and 200 μg/mL streptomycin. Hamsters were observed daily for clinical signs of disease.

**Airborne transmission experiments**. Airborne transmission was examined by co-housing hamsters (1:1) in specially designed cages with a 3D-printed perforated plastic divider dividing the living space in half (Precision Plastics, Inc., Supplementary Fig. 3). This divider prevented direct contact between the donor/primary infected and sentinel hamster and the movement of bedding material. Regular bedding was replaced by alpha-dri bedding to avoid the generation of dust particles. Donor hamsters were infected intranasally as described above and sentinel hamsters placed on the other side of a divider afterwards. Hamsters were followed as described above until 21 DPI. Experiments were performed with cages placed

into a standard rodent cage rack, under normal airflow conditions (Fig. 6c–e). Sentinels were either placed in the direction of the airflow, or against it (Fig. 6b).

**Fomite transmission experiments.** Fomite transmission was examined by infecting donor hamsters as described above by I.N. inoculation. Two animals per cage were housed for 4 days. Regular bedding was replaced by alpha-dri bedding to avoid the generation of dust particles. At 4 DPI, donors were euthanized, and sentinel animals (2 animals per cage) were placed into the contaminated cage (Fig. 5a). Hamsters were followed as described above until DPI 21; bedding and cages were left undisturbed.

**Particle sizing.** Transmission cages were modified by introducing an inlet on the side of the infected hamster side, and sample ports on each end of the cage for measurement of particles in the air under constant airflow condition. Particles were generated by spraying a 20% (v/v) glycerol solution with a standard spray bottle through the cage inlet. The particle size range of the generated particles was measured using a Model 3321 aerodynamic particle sizer spectrometer (TSI). The cage was coated with two sprays at an interval of 30 seconds (s) and after a third spray the sample port was opened, and a sample was analyzed. The cage was sprayed every 30 s and five samples were analyzed (5 runs, each 60 s) for both donor side (primary infected side) and sentinel side.

**Histopathology and immunohistochemistry.** Necropsies and tissue sampling were performed according to IBC-approved protocols. Tissues were fixed for a minimum of 7 days in 10% neutral buffered formalin with 2 changes. Tissues were placed in cassettes and processed with a Sakura VIP-6 Tissue Tek, on a 12-hour automated schedule, using a graded series of ethanol, xylene, and ParaPlast Extra. Prior to staining, embedded tissues were sectioned at 5 μm and dried overnight at 42 °C. Using GenScript U864YFA140-4/CB2093 NP-1 (1:1000) specific anti-CoV immunoreactivity was detected using the Vector Laboratories ImPress VR anti-rabbit IgG polymer (# MP-6401) as secondary antibody. The tissues were then processed using the Discovery Ultra automated processor (Ventana Medical Systems) with a ChromoMap DAB kit Roche Tissue Diagnostics (#760-159).

**Viral RNA detection.** Swabs from hamsters were collected as described above. Cage and bedding material were sampled with prewetted swabs in 1 mL of DMEM supplemented with 200 U/mL penicillin and 200 μg/mL streptomycin. Then, 140 μL was utilized for RNA extraction using the QIAamp Viral RNA Kit (Qiagen) using QIAcube HT automated system (Qiagen) according to the manufacturer's instructions with an elution volume of 150 μL. Sub-genomic (sg) viral RNA and genomic (g) was detected by qRT-PCR[27,68] (Supplementary Table 3). Five μL RNA was tested with TaqMan™ Fast Virus One-Step Master Mix (Applied Biosystems) using QuantStudio 6 Flex Real-Time PCR System (Applied Biosystems) according to instructions of the manufacturer. Ten-fold dilutions of SARS-CoV-2 standards with known copy numbers were used to construct a standard curve and calculate copy numbers/mL.

**Viral titration.** Viable virus in tissue samples was determined as follows. In brief, lung, trachea, brain, and gastrointestinal tissue samples were weighted, then homogenized in 1 mL of DMEM (2% FBS). VeroE6 cells were inoculated with ten-fold serial dilutions of tissue homogenate, incubated 1 h at 37 °C, the first two dilutions washed twice with 2% DMEM. Cells were incubated with tissue homogenate for 6 days, then scored for cytopathic effect. $TCID_{50}$/mL was calculated by the method of Spearman-Karber and adjusted for tissue weight[69].

**Serology.** Serum samples were inactivated with γ-irradiation (2 mRad) and analyzed as follows. In brief, maxisorp plates (Nunc) were coated with 50 ng spike protein (generated in-house) per well and incubated overnight at 4 °C. After blocking with casein in phosphate buffered saline (PBS) (ThermoFisher) for 1 h at room temperature (RT), serially diluted 2-fold serum samples (duplicate, in blocking buffer) were incubated for 1 h at RT. Spike-specific antibodies were detected with goat anti-hamster IgG Fc (horseradish peroxidase (HRP)-conjugated, Abcam) for 1 h at RT and visualized with KPL TMB 2-component peroxidase substrate kit (SeraCare, 5120-0047). The reaction was stopped with KPL stop solution (Seracare) and read at 450 nm. Plates were washed 3 to 5 x with PBS-T (0.1% Tween) for each wash. The threshold for positivity was calculated as the average plus 3 x the standard deviation of negative control hamster sera[70].

**Cytokine analysis.** Cytokine concentrations were determined using a commercial hamster ELISA kit for TNF-α, INF-γ, IL-6, IL-4, and IL-10 available at antibodies.com, according to the manufacturer's instructions (antibodies.com; A74292, A74590, A74291, A74027, A75096). Samples were pre-diluted 1:50.

**Next-generation sequencing of lung mRNA.** Frozen tissues were pulverized in 1 mL of Trizol (Thermofisher Scientific, Waltham, MA), 200 μL of 1-Bromo-3-chloropropane (MilliporeSigma, St. Louis, MO) was added, samples mixed, and centrifuged at 16,000×g for 15 min at 4 °C. RNA containing aqueous phase of 600 μL was collected from each sample and passed through Qiashredder column

(Qiagen, Valencia, CA) at 21,000 x g for 2 minutes to homogenize any remaining genomic DNA in the aqueous phase. Aqueous phase was combined with 600uL of RLT lysis buffer (Qiagen, Valencia, CA) with 1% beta mercaptoethanol (MilliporeSigma, St. Louis, MO) and RNA was extracted using Qiagen AllPrep DNA/RNA 96-well system (Valencia, CA). An additional on-column DNase 1 treatment was performed during RNA extraction. RNA was quantitated by spectrophotometry and yield ranged from 0.4 to 17.8 μg. One hundred nanograms of RNA was used as input for rRNA depletion and NGS library preparation following the Illumina Stranded Total RNA Prep Ligation with Ribo-Zero Plus workflow (Illumina, San Diego, CA). The NGS libraries were prepared, amplified for 13 cycles, AMPureXP bead (Beckman Coulter, Brea, CA) purified using 0.95X beads, assessed on a BioAnalyzer DNA1000 chip (Agilent Technologies, Santa Clara, CA) and quantified using the Kapa Quantification Kit for Illumina Sequencing (Roche, Basel, Switzerland). Amplified libraries were pooled at equal molar amounts and sequenced on a NextSeq (Illumina) using two High Output 150 cycle chemistry kits. Raw fastq reads were trimmed of Illumina adapter sequences using cutadapt version 1.12 and then trimmed and filtered for quality using the FASTX-Toolkit (Hannon Lab). Remaining reads were aligned to the Mesocricetus auratus genome assembly version 1.0 using Hisat2[71]. Reads mapping to genes were counted using htseq-count[72]. Differential expression analysis was performed using the Bioconductor package DESeq2[73]. Pathway analysis was performed using Ingenuity Pathway Analysis (QIAGEN) and gene clustering was performed using Partek Genomics Suite (Partek Inc., St. Louis, MO). Samples with too low quality were removed from the analysis (Supplementary Table 1).

**Next-generation sequencing of virus.** For sequencing from viral stocks, sequencing libraries were prepared using Stranded Total RNA Prep Ligation with Ribo-Zero Plus kit per manufacturer's protocol (Illumina) and sequenced on an Illumina MiSeq at 2 ×150 base pair reads. For sequencing from swab and lung tissue, total RNA was depleted of ribosomal RNA using the Ribo-Zero Gold rRNA Removal kit (Illumina). Sequencing libraries were constructed using the KAPA RNA HyperPrep kit following manufacturer's protocol (Roche Sequencing Solutions). To enrich for SARS-CoV-2 sequence, libraries were hybridized to myBaits Expert Virus biotinylated oligonucleotide baits following the manufacturer's manual, version 4.01 (Arbor Biosciences, Ann Arbor, MI). Enriched libraries were sequenced on the Illumina MiSeq instrument as paired-end 2 ×150 base pair reads. Raw fastq reads were trimmed of Illumina adapter sequences using cutadapt version 1.1227 and then trimmed and filtered for quality using the FASTX-Toolkit (Hannon Lab, CSHL). Remaining reads were mapped to the SARS-CoV-2 2019-nCoV/USA-WA1/2020 genome (MN985325.1) using Bowtie2 version 2.2.928 with parameters-local-no-mixed -X 1500. PCR duplicates were removed using picard MarkDuplicates (Broad Institute) and variants were called using GATK HaplotypeCaller version 4.1.2.029 with parameter -ploidy 2. Variants were filtered for QUAL > 500 and DP > 20 using bcftools.

**Statistical analysis.** Heatmaps and correlation graphs were made in R[74] using pheatmap[75] and corrplot[76] packages. Significance test were performed as indicated where appropriate: Spearman correlation test, two-way ANOVA and Kruskal–Wallis test. Statistical significance levels were determined as follows: ns = $p > 0.05$; *$p \le 0.05$; **$p \le 0.01$; ***$p \le 0.001$; ****$p \le 0.0001$.

**Reporting summary.** Further information on research design is available in the Nature Research Reporting Summary linked to this article.

## Data availability
The data generated in this study have been deposited in the figshare repository (https://doi.org/10.6084/m9.figshare.14642502). RNA-Seq data was deposited in NCBI GEO under accession number GSE177027. Source data is provided with this paper. Source data are provided with this paper.

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

## Acknowledgements

The authors would like to thank the Rocky Mountain Veterinary branch, including Nicki Arndt, Amanda Weidow and Brian Mosbrucker for assistance with high containment husbandry and cage design and testing, Greg Saturday for assistance with necropsy, Tina Thomas for assistance with histology, Stephanie Seifert for assistance in study protocol editing, Kimmo Virtaneva and Stacy Ricklefs for assistance with sequencing and Rose Perry and Ryan Kissinger for assistance with the figures. This research was supported by the Intramural Research Program of the National Institute of Allergy and Infectious Diseases (NIAID), National Institutes of Health (NIH).

## Author contributions

Conceptualization: J.R.P. and C.K.Y.; methodology: J.R.P., C.K.Y., C.S.C., I.O.O., M.H., R.F., T.B., J.E.S. and C.M.; resources: V.A.A. and N.v.D.; supervision: V.J.M.; data curation: J.R.P., C.S.C. and C.K.Y.; writing: J.R.P., C.K.Y. and V.J.M.; visualization: J.R.P. and C.K.Y.

## Competing interests

The authors declare no competing interests.
