## [Peer Review File · Nature Communications]

Reviewers' Comments:

Reviewer #1:

Remarks to the Author:

This paper uses the Syrian hamster model to investigate four "natural" routes of SARS-CoV-2 infection and the impact of these routes on disease severity, kinetics, transmissibility, and immune response. The routes are fomite, intranasal, airborne, and aerosol. One notable finding was that routes resulting in upper respiratory infection (fomite and IN) tended to exhibit more shedding but less severe lung disease than aerosol exposure. Other interesting observations were that fomite exposure appeared to cause less severe disease, but nevertheless essentially all of the animals seroconverted, albeit at lower levels than IN or aerosol exposure. Another interesting finding was that more exposure to the lung (aerosol) resulted in more severe disease but not more shedding (relative to IN and fomite). There were some preliminary experiments looking at airborne transmission between cages that appear to be precursors to more rigorous future experiments.

Issues: Only one dose was used for the different routes. A dilution series would be need to assess the effects of dose via each route. Otherwise, the difference in "route" observed could actually all be attributed to actual dose of exposure (this is especially true for the IN and fomite comparison). T It seems likely that the dose of virus by fomite exposure was very low. It isn't known. The inhaled dose by aerosol was calculated but I am not sure if proper sampling was made to confirm the aerosol dose (all-glass impinge, AGI, measures). Regardless of those measures, a dose response by the different routes was not evaluated and compared. Would the same results be observed if the concentration of virus in the fomite was 10 or 100 times higher, or if the IN or aerosol was 10 to 100 times lower? For example, if IN or aerosol dosing were reduced, would we see seroconversion without overt disease (asymptomatic)?

It was unclear to me if this is the first time the aerosol route has been used to expose hamsters to SARS2. If it is, that should be emphasized.

The title states that "...severity and transmission efficiency is increased for airborne but not fomite exposure," but it does not state what that is relative to. Increased relative to what? Relative to IN? Same issue in abstract (lines 24-25). We need to know what that increase is relative to. In that sentence the increase might be relative to fomite...

Line 269. A better description of this cage divider should be given in the Methods since there are airflow-related results that could be affected by the number of perforations etc.

There are several sentences discussing the importance of direction of airflow on airborne transmission in the results and Discussion. Perhaps the authors can say something like, "this confirms what seems obvious, but it provides experimental evidence supporting the importance for control measures focused on strategically designed room vent etc....."

Line 308. Consider including "dose" along with route. "... relationship between exposure route, dose, and the differential impact...."

Line 342. Most masks are designed to prevent inhalation of large droplets into the upper respiratory, but they do not prevent aerosols (Gandhi, 2020). Thus they would not prevent aerosols from deposition in the lower respiratory tract. If deposition to upper respiratory can lead to more asymptomatic cases, and deposition to the lower respiratory tract can lead to more severe disease... does that have implications for the types of masks people are wearing?

Line 426. How long was the fomite pan in the cage? This could be added to method.

Minor issues:

- Line 30. Remove "of"
- Line 40 awkward sentence
- Is "inoculate" the right word to use when not referring to vaccination?
- Line 236- Section heading "may predict." Speculation in the Results section should be in the Discussion section.
- Line 284. Clarify "direct inoculation" direct IN inoculation?

Reviewer #2:

Remarks to the Author:

The manuscript NCOMMS-21-00578-T entitled "Natural Routes of Infection Determine Transmission Dynamics and Disease Manifestation of SARS-CoV-2 in the Syrian Golden Hamster" by Port J.R. et al focuses on how the route of infection affects the disease severity and transmissibility. The authors use a well-defined hamster model of SARS-CoV-2 to address the effect of different exposure routes, either intranasal, aerosol or fomite and address both the effect on the exposed animal as well as the transmissibility to naive animals in contact. The focus of the manuscript is of major importance and the overall data supports the questions raised by the authors. The data is mostly novel and the authors adequately discuss previous achievements in the field (Sia S.F. Nature 2020). Methods and statistical analyses are appropriate, the manuscript is clearly written. The results are mostly well presented, and required data is listed below.

Several major issues need to be addressed:

1. The risk of infection through fomite exposure is discussed, yet, the authors should discuss additional manuscripts (e.g. Ben-Shmuel A. et al. <https://doi.org/10.1016/j.cmi.2020.09.004>) describing inefficient isolation of culturable viruses from surfaces and air filters despite the stability of the virus in laboratory conditions. Data on contaminated surfaces as a source of fomite infection in humans needs to be discussed and compared to the hamsters model and the effect of decontamination agents and procedures used in hospitals should be considered in addition to the distance from the source of fomites to the exposed respiratory that is significantly different between humans and hamsters.
 2. The authors need to describe clearly how the exposure dose was determined in aerosol and fomite exposure experiments, whether the value is an estimated number/dose applied to the plate/aerosol generator or whether the aerosol or fomite was collected and titrated.
 3. Fig. 1 and 2 – While the authors show staining of viral antigens in the nasal turbinate (fig.2) the viral load was only measured at the trachea (Fig 1). This needs to be addressed as the nasal turbinate is efficiently infected and can serve as a source for virus dissemination.
 4. Also, in Fig.1d virus is detected in the trachea following intranasal infection, yet no staining is observed in fig. 2. This needs to be explained/solved.
 5. Description of specific cell types (lines 108, 140) is not supported by specific staining or higher magnifications that would support the description. Please provide supportive data.
 6. As the authors conclude, fomite exposure results in a delayed disease. Thus, the effect of fomite exposure on the target organs should have been determined also at later time points (e.g. day 7 or 8).
 7. Analysis of the inflammatory immune response (Fig. 4) is the weakest point in the manuscript. The data is very weak and the significance is low. Also, Fig. 4a is the TNF alpha following IN and aerosol exposure is significantly different from unexposed? If not what is the meaning. Fig. 4a IL10 and IL4 – differ in response to the exposure routes. Why? As cytokine profiling in hamsters may not be easy to address at the protein level, the authors may consider strengthening the data with RNA analyses or by removing the data from the manuscript body. Currently, this data hampers the strength of this important work.
 8. Lines 203-206 – the data is weak and only binding antibodies are shown. The authors should provide at least neutralization data to support their conclusion of protective immunity. Also, it can be concluded that fomite exposure is characterized by delayed immune response rather than to rely on weak data to support a mechanism of anti-inflammatory response.
 9. Line 246, the intranasal data does not support the conclusion as early shedding is not followed by acute manifestation as compared to aerosol.
 10. Lines 345-353, the secretion was measured through sgRNA but culturable virus was not determined and co-caging was only addressed following intranasal instillation. The conclusions should be adjusted.
- Finally, in light of the novel SARS-CoV-2 variants that strongly affect the globe, it would be of major importance at least to discuss their possible effect on virus transmissibility and whether their unique features are expected to affect the route of transmission.

Reviewed by: Nir Paran

Reviewer #3:

Remarks to the Author:

The report by Port et al claims that route and form of infection with SARS-CoV-2 influences the severity of disease and kinetics of viral shedding. This is a very important and timely study and their finding could have impact on public health measures to the COVID-19 pandemic.

There has been much debate about the mechanism of infection, especially in household contacts, regarding infection of household contacts as in such settings there is generally very little evidence of high levels of SARS-2 in fomites. Furthermore, the suggestion that face coverings can reduce the level of SARS-CoV-2 aerosols reaching the lower respiratory tract is also an extremely important finding. As the authors highlight, the models described could also be applied to assessing the phenotype of the many VOCs that are continually evolving.

The authors have also chosen the most appropriate in vivo model to perform their studies. The hamster has been shown by numerous groups to have many advantages over other species for the study of SARS-CoV-2.

In general, I find that the experimental design and execution adequate to support the authors claims throughout the manuscript. Though, like many COVID-19 research studies, larger animal groups would have been preferable for several the parameters measured. This is especially the case when trying to make conclusions when $n=2$! The statistical analysis applied throughout the study also appears appropriate.

Though the studies appear appropriate and well executed, the key parameter that does not appear to be fully characterised is the quality of the SARS-CoV-2 challenge stock used throughout the paper. The authors state the nCoV-WA1-2020 isolate was propagated in VERO E6 cells.

Unfortunately, it has been widely reported that expansion in this cell line will rapidly promote the growth of an 8 aa furin cleavage site variant (A Davidson et al 2020). It has also been suggested that significant changes in the viral spike protein could have an impact of virus phenotype which may have influenced the findings in this manuscript. I would at least like to see the full sequence characterisation of the challenge stock, including major populations of minor variants in addition to the consensus sequence. Furthermore, the particle to infectivity ration could also dictate how a pathogen behaves in vivo. This could simply be calculated for the challenge stock and compared to the P:I ratio of the virus shed by the hamster as the authors have already calculated the genome copy number and TCID50 of such samples.

Recognition of the different characteristics of a VERO E6 cultivated virus prep compared to naturally shed virus should be highlighted throughout the manuscript and especially in the discussion when extrapolating this data to the clinical setting.

On a more minor note, can the authors confirm it the wire cage holding device presents a nose only or whole-body delivery of aerosol. If the latter, how do they think this would influence the subsequent exposure of the animal by fomites that will have formed on their fur?

Can the authors also highlight if severity of disease was related to potency of immunity in the hamsters?

REVIEWER COMMENTS

Reviewer #1 (Remarks to the Author):

This paper uses the Syrian hamster model to investigate four “natural” routes of SARS-CoV-2 infection and the impact of these routes on disease severity, kinetics, transmissibility, and immune response. The routes are fomite, intranasal, airborne, and aerosol. One notable finding was that routes resulting in upper respiratory infection (fomite and IN) tended to exhibit more shedding but less severe lung disease than aerosol exposure. Other interesting observations were that fomite exposure appeared to cause less severe disease, but nevertheless essentially all of the animals seroconverted, albeit at lower levels than IN or aerosol exposure. Another interesting finding was that more exposure to the lung (aerosol) resulted in more severe disease but not more shedding (relative to IN and fomite). There were some preliminary experiments looking at airborne transmission between cages that appear to be precursors to more rigorous future experiments.

Issues:

Only one dose was used for the different routes. A dilution series would be needed to assess the effects of dose via each route. Otherwise, the difference in “route” observed could actually all be attributed to actual dose of exposure (this is especially true for the IN and fomite comparison). It seems likely that the dose of virus by fomite exposure was very low. It isn't known. The inhaled dose by aerosol was calculated but I am not sure if proper sampling was made to confirm the aerosol dose (all-glass impinge, AGI, measures). Regardless of those measures, a dose response by the different routes was not evaluated and compared. Would the same results be observed if the concentration of virus in the fomite was 10 or 100 times higher, or if the IN or aerosol was 10 to 100 times lower? For example, if IN or aerosol dosing were reduced, would we see seroconversion without overt disease (asymptomatic)?

We agree with the reviewer that a dose response investigation across different exposure routes would be of interest. However, our aim in this study was to compare cross-sectionally when animals are presented with a similar dosage across the routes. In one of our previous papers, Rosenke et al, 2020, we showed that increased infectious dose does not affect shedding or disease severity in the Syrian hamster. In addition, in preliminary, unpublished

data from our group we demonstrate that intranasal inoculation even with 10 TCID_{50} has led to similar weight loss and lung pathology. For IN, this suggest that an infection, independent of dose, results in a comparable disease kinetics.

As we do not see the same weight loss or disease pathology after fomite exposure, we draw the conclusion that the differences observed in this paper are exposure route dependent and not a result of only a difference in dosage. It is beyond the aim of this paper to address the dose-dependency for each individual route, as we aim here to provide a first direct comparison between the routes when animals are exposed to comparable amounts of virus. We have addressed in the relevant sections of the manuscript that the aerosol dose was confirmed by sampling of the air and performing a back titration. Similarly, the virus amount provided by fomite was confirmed. We acknowledge that it remains practically impossible, however, to truly address the amount of virus that the individual hamster will be exposed to in this route. As such, we have addressed this issue in the discussion as follows:

Line 345: “Our data suggest that in addition to the exposure dose [29] and underlying host conditions [30], disease is a function also of exposure route. “

Line 391: “Additionally, while dosage of I.N. and aerosol exposure could be accurately determined, the caveat must be given, that the exact dosage through fomite could not be confirmed. While previous work in this model has demonstrated that disease severity and shedding profile is not overtly affected by infection dose after I.N. exposure [24], we cannot confirm that the dose-dependency after aerosol and fomite exposure may not present differently.”

It was unclear to me if this is the first time the aerosol route has been used to expose hamsters to SARS2. If it is, that should be emphasized.

This is the first time, to the best of our knowledge, aerosol route is used to expose hamsters. We have added this information to the discussion section of the manuscript in this sentence:

Line 352: “Here we directly compared natural transmission routes, designed to mimic airborne and fomite exposure by presenting the first data on aerosol and direct fomite inoculation in this model.”

The title states that “...severity and transmission efficiency is increased for airborne but not fomite exposure,” but it does not state what that is relative to. Increased relative to what? Relative to IN? Same issue in abstract (lines 24-25). We need to know what that increase is relative to. In that sentence the increase might be relative to fomite...

We agree with the reviewer that this statement needs additional clarification. We are directly comparing the routes with each other. Therefor modified the title to:

“SARS-CoV-2 disease severity and transmission efficiency is increased for airborne compared to fomite exposure in Syrian hamsters”

In addition, we modified the abstract in line 25 to

“Intranasal and aerosol inoculation caused more severe respiratory pathology, higher virus loads and increased weight loss. In contrast, fomite exposure led to milder disease manifestation characterized by an anti-inflammatory immune state and delayed shedding pattern.”

Line 269. A better description of this cage divider should be given in the Methods since there are airflow-related results that could be affected by the number of perforations etc.

We agree with the reviewer and have added additional information on the cage divider in the method section. In addition, we have included a very detailed construction figure of the divider in supplement. (Line 521, Sup Fig 3)

There are several sentences discussing the importance of direction of airflow on airborne transmission in the results and Discussion. Perhaps the authors can say something like, “this confirms what seems obvious, but it provides experimental evidence supporting the importance for control measures focused on strategically designed room vent etc.....”

On the suggestion of the reviewer we have now added this in our discussion section and it now reads:

Line 449: “The results of the experiments with directional and reverse-directional airflow provide direct experimental data supporting preemptive SARS-CoV-2 control measures focused on improvement of ventilation.”

Line 308. Consider including “dose” along with route. “... relationship between exposure route, dose, and the differential impact...”

We have considered the reviewer’s comment and have modified the sentence as suggested. (now line 339)

Line 342. Most masks are designed to prevent inhalation of large droplets into the upper respiratory, but they do not prevent aerosols (Gandhi, 2020). Thus, they would not prevent aerosols from deposition in the lower respiratory tract. If deposition to upper respiratory can lead to more asymptomatic cases, and deposition to the lower respiratory tract can lead to more severe disease... does that have implications for the types of masks people are wearing?

Currently the exact mechanism of filtration by different types of masks is still unclear. While

certain mask appears to be largely focused on reduction of larger droplets, electrostatic filtration appears to result in a reduction of smaller particles as well. We are currently designing experiments to study exactly the question posed by the reviewer.

Line 426. How long was the fomite pan in the cage? This could be added to method. **It was in the cage for one day. We have added this to the methods (now line 507). We have also added the fact that interaction of hamsters with the dish was monitored and confirmed within the first 5 minutes after placing it into the cage.**

Minor issues:

- Line 30. Remove “of”

We have removed this.

- Line 40 awkward sentence

We have modified this sentence as follows:

Line 24: “Intranasal and aerosol inoculation caused severe respiratory pathology, higher virus loads and increased weight loss.”

- Is "inoculate" the right word to use when not referring to vaccination?

Yes, we think that inoculate is more appropriate. We have confirmed this in the merriam and webster dictionary and one of the definitions is "to introduce a microorganism into"

- Line 236- Section heading “may predict.” Speculation in the Results section should be in the Discussion section.

We have removed the word “may” from the sentence. (line 267)

- Line 284. Clarify “direct inoculation” direct IN inoculation?

We have clarified this to direct IN inoculation. (line 316)

Reviewer #2 (Remarks to the Author):

The manuscript NCOMMS-21-00578-T entitled “Natural Routes of Infection Determine Transmission Dynamics and Disease Manifestation of SARS-CoV-2 in the Syrian Golden Hamster” by Port J.R. et al focuses on how the route of infection affects the disease severity and transmissibility. The authors use a well-defined hamster model of SARS-CoV-2 to address the effect of different exposure routes, either intranasal. Aerosol or fomite and address both the effect on the exposed animal as well as the transmissibility to naïve animals in contact.

The focus of the manuscript is of major importance and the overall data supports the questions raised by the authors. The data is mostly novel, and the authors adequately discuss previous achievements in the field (Sia S.F. Nature 2020). Methods and statistical analyses are appropriate, the manuscript is clearly written. The results are mostly well presented, and required data is listed below.

Several major issues needs to be addressed:

1. The risk of infection through fomite exposure is discussed, yet, the authors should discuss additional manuscripts (e.g Ben-Shmuel A. et al. <https://doi.org/10.1016/j.cmi.2020.09.004>) describing inefficient isolation of culturable viruses from surfaces and air filters despite the stability of the virus in laboratory conditions. Data on contaminated surfaces as a source of fomite infection in humans needs to be discussed and compared to the hamsters model and the effect of decontamination agents and procedures used in hospitals should be considered in addition to the distance from the source of fomites to the exposed respiratory that is significantly different between humans and hamsters.

This is an accurate observation made by the reviewer and we have included a section in the discussion to address these suggestions:

Line 423: “There has been considerable effort to culture virus samples from contaminated surfaces or air, yet there is significant discrepancy between viral RNA detected and actual isolation of live virus [8, 14, 50]. This may be due to a lack of efficient culture methods or a result of overestimation of the viral environmental burden based on PCR methodology. As we assessed the environmental contamination in hamster cages by PCR and found it consistently high even after shedding of infectious virus by the donors is expected to have ceased [25], this may suggest a transmission risk even when culturable virus may not be found. Additionally, it needs to be acknowledged, that the hamster and human interaction with potential fomites is not equal. While hamsters can be assumed to interact with their environment in more intimate and consistent manner, the risk of fomite exposure demonstrated in this study does highlight the important role of reducing tactile interactions with potentially contaminated surfaces also for humans. Further, it does support countermeasures such as hand-washing and regular de-contamination of surfaces, which are likely to reduce the risk of infection [51].”

2. The authors need to describe clearly how the exposure dose was determined in aerosol and fomite exposure experiments, whether the value is an estimated number/dose applied to the plate/aerosol generator or whether the aerosol or fomite was collected and titrated.

We agree with the reviewer that the wording in the M&M was a bit unclear. The doses of the aerosol inoculations were calculated using infectious titers in the aerosols to which the animals were exposed. We have added this information in the manuscript.

Line 498: “A sample of 6 liters of air per min was collected during the 10min exposure on the 47mm gelatin filter. Post exposure, the filters were dissolved in 10 mL of DMEM containing 10% FBS and infectious virus was titrated as described above and the aerosol concentration was calculated. The estimated inhaled inoculum was calculated using the respiratory minute volume rates of the animals determined using the methods of Alexander *et al.* [66].”

In addition, we added language on the fomite exposure as well.

Line 506: “Fomite exposure was conducted by placing a polypropylene dish into the cage containing 40µL of 8×10^4 TCID₅₀ SARS-CoV-2 per hamster (total dose per cage: 3.2×10^5 TCID₅₀) for 24 h. Interaction of hamsters with the dish was monitored and confirmed within the first 5 minutes after placing it into the cage. For I.N. and fomite exposure undiluted stock virus with confirmed dose was applied.”

3. Fig. 1 and 2 – While the authors show staining of viral antigens in the nasal turbinate (fig.2) the viral load was only measured at the trachea (Fig 1). This needs to be addressed as the nasal turbinate is efficiently infected and can serve as a source for virus dissemination.

The reviewer is correct. Unfortunately, to preserve the integrity of the nasal turbinates for histological analysis, we were unable to preserve samples for virus titrations. However, we do feel that the orals swabs are a good proxy for the spatial and temporal dynamics of SARS-CoV-2 infection in hamsters.

4. Also, in Fig.1d virus is detected in the trachea following intranasal infection, yet no staining is observed in fig. 2. This needs to be explained/solved.

We have addressed this discrepancy in the discussion section as follows:

Line 356: “While SARS-CoV-2 RNA was detected by PCR of tracheal tissue at 1 DPI from I.N. inoculated animals, SARS-CoV-2 N protein antigen was not detected by IHC at this timepoint. Several factors may contribute to this: proximal trachea was taken for molecular and virologic analysis, which is the site in closest physical proximity to the site of inoculation. This site may have been infected more rapidly due to the proximity to the inoculation dose. Additionally, IHC can be a highly specific assay but may be poorly sensitive during the pre-acute phase of disease when little viral antigen is being produced.”

5. Description of specific cell types (lines 108, 140) is not supported by specific staining or higher magnifications that would support the description. Please provide supportive data.

Upon the suggestion of the reviewer we have now included a figure with a higher magnification which is now included in the supplement as Sup Fig 1b.

6. As the authors conclude, fomite exposure results in a delayed disease. Thus, the effect of fomite exposure on the target organs should have been determined also at later time points (e.g. day 7 or 8).

We acknowledge the comment of the reviewer. We observed delayed viral replication and not disease, we have adjusted this in the manuscript accordingly:

In the results section:

Line 176: “Fomite SARS-CoV-2 exposure results in a reduced immune profile in the lung”

And in the discussion section:

Line 365: “In contrast, fomite inoculation displayed a prolonged time between exposure and viral replication in the lung leading to reduced disease severity.”

The observed weight kinetics in the fomite animals do not suggest a delayed pathology at 7 or 8 DPI, and as such we do not see the added benefit of including a new study at day 7 and/or 8.

7. Analysis of the inflammatory immune response (Fig. 4) is the weakest point in the manuscript. The data is very weak, and the significance is low. Also, Fig. 4a is the TNF alpha following IN and aerosol exposure is significantly different from unexposed? If not, what is the meaning. Fig. 4a IL10 and IL4 – differ in response to the exposure routes. Why? As cytokine profiling in hamsters may not be easy to address at the protein level, the authors may consider strengthening the data with RNA analyses or by removing the data from the manuscript body. Currently, this data hampers the strength of this important work.

We acknowledge, that the provided analysis of the serum cytokine response is not as in-depth and detailed to what could be achieved in a study with more conventional laboratory species such as mice or were the focus of the study is more immunology and host-response oriented. This largely due to a lack of hamster-specific reagents. However, we respectfully disagree with the reviewer that the conclusions drawn are weak and we think that our conclusions are still supported by the data.

We have modified the relevant section to address the decreased TNF-alpha response in fomite animals, not an increase in aerosol and I.N. groups.

Line 178: “While overall significance was low, serum levels were different depending on exposure route for pro-inflammatory tumour necrosis factor (TNF)- α and anti-inflammatory interleukin (IL)-4 and IL-10. In contrast to unexposed, I.N. and aerosol groups, fomite exposed animals presented with decreased levels of TNF- α at 4 DPI;”

The data does indeed show that fomite exposed animals mount an anti-inflammatory response characterize by decreased TNF-alpha, but increased IL-10. The reviewer is correct in pointing out the difference in IL-4 and IL-10 response and we have now addressed both cytokines in the discussion.

“The decrease in TNF- α may reduce immune pathology in the lung even with the observed viral titers at 4 DPI not being significantly lower as compared to aerosol inoculation. It has been previously shown that IL-10 and IL-4 may be regulated in a differentiated and infection-route dependent manner, which would explain the differential systemic presence of these cytokines here [34]. However, our systemic analysis of cytokine profiles was severely limited by the availability of hamster-specific reagents and remains superficial.”

To support the initial dataset and the conclusion we have now included analysis of the response on the mRNA level. To this end, we have conducted next-generation Illumina sequencing on the transcriptome of lungs collected at 1 and 4 DPI. We have focused our analysis on 4 DPI. The data can be found in a new results section. It corroborates the findings that the transmission route has impact on the immune response and that fomite exposed animals demonstrated a reduced upregulation of the immune response which was linked to reduced upregulation of the coronavirus pathogenesis pathway. We also addressed these new findings in the results and discussion section.

Results, line 203: “To gain insight into the local immune responses in the lung, we evaluated global changes in the gene expression at 1 and 4 DPI in comparison to control animals. Three lung samples were removed due to quality issues (Sup Table 1). Principal components analysis revealed expected grouping from most conditions with each group

containing their associated replicates. The largest separation was from groups 1 and 4 DPI aerosol samples, followed by less separation between the remaining six conditions. However, within this second cluster of the remaining six conditions, separate ellipses representing two standard deviations can still be viewed as non-intersecting groups, distinct from the controls (Sup Fig 2).

To assess which pathways were differently regulated for each exposure route, the gene expression information was imported into Integrated Pathway Analysis (IPA) software. The results show that in I.N. and aerosol exposed groups over 50 canonical pathways were up- or downregulated significantly in comparison to control animals (p -value < 0.05 , z -score < -2 or > 2); amongst which metabolic, immune, infection and cell function associated pathways (Fig 4 d, Sup Table 2 shows all significant pathways). In fomite animals, only 10 pathways were found to be significantly up- or downregulated as compared to control animals. In I.N. and aerosol exposed animals, pathway analysis revealed macrophage activation, dendritic cell maturation, interferon signalling and T-, B- and NK-cell involvement. In fomite exposed animals the interferon signalling, Th17 pathway and pattern recognition for bacteria and viruses were upregulated. Interestingly, involvement of the Th17 pathway were found in all three.

As we saw similar virus titers in the lungs of animals at 4 DPI, using IPA we next compared specifically the virus-induced response (coronavirus pathway) and the resulting adaptive immune response (Th1/Th2 pathway) in more detail (Fig 4 e). Aerosol and I.N. exposed animals showed differential expression and regulation of genes associated with the coronavirus pathway. In comparison to I.N. exposed animals, aerosol exposure at 4 DPI was linked to downregulation of multiple key mediators including MAVS, ELK1, BCL2, Serpine 1 and IFNAR1, but upregulation of IL-1 β . In comparison, we found no major differential expression of the coronavirus pathway associated genes in fomite exposed animals. In detail, the Th1/Th2 pathway upregulation comprised in I.N. and aerosol animals amongst others increased gene expression of pro-inflammatory cytokines IL-18, IFN- γ , IL-6 and IL-2, upregulation of expression of surface molecules CD4 and CD8, multiple co-activation molecules like CD28, CD80, CD40 and chemokine receptors and tissue trafficking receptors such as CXCR3, CXCR6, CCR8 and ITGB2. In contrast, fomite exposed animals showed minimal upregulation and clustered closest to controls. Taken together this suggests a predominantly mild immune response, as compared to aerosol exposure, is mounted after fomite exposure in the lung which may protect from more severe outcome.“

Discussion, line 375: “Our data on the local transcriptome at 4 DPI demonstrates that aerosol and I.N. exposure led to major and differential involvement of coronavirus pathway associated genes and increased the upregulation of a subset of genes in the Th1 and Th2 response pathways, whereas fomite exposure resulted in minimal pathway activation at this time point. This could further explain, why in these animals we observed increased pathology, which is absent in the fomite animals. Previously, single-cell analysis in the hamster has demonstrated that inflammation and CD4 $^+$ and CD8 $^+$ cytotoxic T-cell responses preceded viral elimination in the hamster [35]. The gene expression profile seen after aerosol and I.N. exposure would also suggest increased T-cell, NK cell and macrophage recruitment to the site of infection, as well as activation of the humoral response. Interestingly fomite exposed animals still mounted a considerable humoral response, which could further imply that the immune response in these animals may be driven by infection outside of the lung. If these differential immune-signatures are not only a result of dose-kinetic but an intrinsic effect of the exposure routes, this could have important implications in the context of re-infection with novel variants.”

8. Lines 203-206 – the data is weak and only binding antibodies are shown. The authors should provide at least neutralization data to support their conclusion of protective immunity. Also, it can be concluded that fomite exposure is characterized by delayed immune response rather than to rely on weak data to support a mechanism of anti-inflammatory response.

We have included the VN data for the different routes which supports the fact that fomite exposure leads to a weaker, but functional, humoral immune response. This data has been added to results section and figure 4 has been updated.

Line 198: “We compared the neutralizing capacity against live virus. Aerosol exposed animals demonstrated highest neutralizing titers and fomite exposed animals lowest, however not to significant difference (N = 4, Kruskal-Wallis test, followed by Dunn’s multiple comparisons test, $p = 0.2026$), and the ratio between neutralizing titers and ELISA titers did equally show no difference (Sup Fig 1 e).”

Line 282: “Interestingly, olfactory pathology also showed positive correlation to the magnitude of the IgG response and the neutralizing capacity (Spearman correlation test, N = 12, $p = 0.001$ and $p = 0.021$, respectively) (Fig 5 e).”

9. Line 246, the intranasal data does not support the conclusion as early shedding is not followed by acute manifestation as compared to aerosol.

We are not completely sure about the reviewer’s comment here. We noticed that both aerosol and intranasal exposure have early rectal shedding at day 2, more TNF-alpha in the serum compared to fomite exposed animals and increased olfactory pathology.

10. Lines 345-353, the secretion was measured through sgRNA but culturable virus was not determined and co-caging was only addressed following intranasal instillation. The conclusions should be adjusted.

We thank the reviewer for this comment. Indeed, we measured contamination of the cages through genomic and sub genomic RNA. At these lines we are discussed the findings of our current study (genomic and subgenomic RNA based) in the context of a previous paper that assessed shedding of infectious virus after IN inoculation with a similar dose (Sia et al., 2020), which was measured by titration. Based on both observation we concluded that environmental contamination is reduced at 4 DPI.

Finally, in light of the novel SARS-CoV-2 variants that strongly affect the globe, it would be of major importance at least to discuss their possible effect on virus transmissibility and whether their unique features are expected to affect the route of transmission.

The reviewer is making a very good point, and we have addressed this in the discussion:

Line 457: “The findings of this study suggest that using more natural routes of transmission is highly suitable for accurately assessing the transmission potential and pathogenicity of novel evolved strains [61]. Novel variants such as B.1.1.7 and B.1.351 are currently replacing the old circulating variants. Shedding patterns in humans suggest increased transmissibility may not only be a function of increased binding capacity to

hACE2 or replication [62-64]. If these variants are shown to have different environmental stability, it would be prudent to also investigate if this impacts exposure routes.”

Reviewed by: Nir Paran

Reviewer #3 (Remarks to the Author):

The report by Port et al claims that route and form of infection with SARS-CoV-2 influences the severity of disease and kinetics of viral shedding. This is a very important and timely study and their finding could have impact on public health measures to the COVID-19 pandemic. There has been much debate about the mechanism of infection, especially in household contacts, regarding infection of household contacts as in such settings there is generally very little evidence of high levels of SARS-2 in fomites. Furthermore, the suggestion that face coverings can reduce the level of SARS-CoV-2 aerosols reaching the lower respiratory tract is also an extremely important finding. As the authors highlight, the models described could also be applied to assessing the phenotype of the many VOCs that are continually evolving. The authors have also chosen the most appropriate in vivo model to perform their studies. The hamster has been shown by numerous groups to have many advantages over other species for the study of SARS-CoV-2.

In general, I find that the experimental design and execution adequate to support the authors claims throughout the manuscript. Though, like many COVID-19 research studies, larger animal groups would have been preferable for several the parameters measured. This is especially the case when trying to make conclusions when n=2! The statistical analysis applied throughout the study also appears appropriate.

Though the studies appear appropriate and well executed, the key parameter that does not appear to be fully characterised is the quality of the SARS-CoV-2 challenge stock used throughout the paper. The authors state the nCoV-WA1-2020 isolate was propagated in VERO E6 cells. Unfortunately, it has been widely reported that expansion in this cell line will rapidly promote the growth of an 8 aa furin cleavage site variant (A Davidson et al 2020). It has also been suggested that significant changes in the viral spike protein could have an impact of virus phenotype which may have influenced the findings in this manuscript. I would at least like to see the full sequence characterisation of the challenge stock, including major populations of minor variants in addition to the consensus sequence.

We acknowledge the reviewer’s concern on the purity and origin of the stock virus used. We did indicate in the methods (as we do in all our manuscripts) that the used virus stock was sequenced and that the consensus sequence of virus stock used was 100% identical to the initial deposited GenBank sequence (MN985325.1) of the WA1 patient sample and no contaminants were detected. No furin cleavage site deletions or other alteration in the spike sequence were observed. We have now added additional detail to the M&M on the deep sequencing of the virus stock.

Line 620: “For sequencing from viral stocks, sequencing libraries were prepared using Stranded Total RNA Prep Ligation with Ribo-Zero Plus kit per manufacturer’s protocol (Illumina) and sequenced on an Illumina MiSeq at 2 x 150 base pair reads. For sequencing from swab and lung tissue, total RNA was depleted of ribosomal RNA using the Ribo-Zero

Gold rRNA Removal kit (Illumina). Sequencing libraries were constructed using the KAPA RNA HyperPrep kit following manufacturer's protocol (Roche Sequencing Solutions). To enrich for SARS-CoV-2 sequence, libraries were hybridized to myBaits Expert Virus biotinylated oligonucleotide baits following the manufacturer's manual, version 4.01 (Arbor Biosciences, Ann Arbor, MI). Enriched libraries were sequenced on the Illumina MiSeq instrument as paired-end 2 X 150 base pair reads. Raw fastq reads were trimmed of Illumina adapter sequences using cutadapt version 1.1227 and then trimmed and filtered for quality using the FASTX-Toolkit (Hannon Lab, CSHL). Remaining reads were mapped to the SARS-CoV-2 2019-nCoV/USA-WA1/2020 genome (MN985325.1 using Bowtie2 version 2.2.928 with parameters --local --no-mixed -X 1500. PCR duplicates were removed using picard MarkDuplicates (Broad Institute) and variants were called using GATK HaplotypeCaller version 4.1.2.029 with parameter -ploidy 2. Variants were filtered for QUAL > 500 and DP > 20 using bcftools.

Furthermore, the particle to infectivity ratio could also dictate how a pathogen behaves in vivo. This could simply be calculated for the challenge stock and compared to the P:I ratio of the virus shed by the hamster as the authors have already calculated the genome copy number and TCID₅₀ of such samples.

The reviewer is correct in his assessment that the P:I ratio between the stock virus and naturally shed virus may be different. Unfortunately, we are not able to accurately calculate the P:I ratio in samples shed from hamsters as TCID₅₀ data is not available from hamster swabs. We have, however, used sgRNA as a surrogate for replicating virus and graphed the g/sgRNA ratio, which can be found now in Supplemental figure 1. We have also added to the manuscript:

Line 388: "We acknowledge that these direct exposure experiments were performed using cultured virus and that it cannot be ruled out that the particle to infectivity ratio may differ from those found in naturally shed samples."

Recognition of the different characteristics of a VERO E6 cultivated virus prep compared to naturally shed virus should be highlighted throughout the manuscript and especially in the discussion when extrapolating this data to the clinical setting.

We have added to discussion the caveat, that comparison of the exposure routes is based on cell-culture grown virus.

Line 388: "We acknowledge that these direct exposure experiments were performed using cultured virus and that it cannot be ruled out that the particle to infectivity ratio may differ from those found in naturally shed samples."

On a more minor note, can the authors confirm if the wire cage holding device presents a nose only or whole-body delivery of aerosol. If the latter, how do they think this would influence the subsequent exposure of the animal by fomites that will have formed on their fur?

This was a full body aerosol exposure. As the reviewer did correctly indicate, fur fomite contamination cannot be completely ruled out. However, given the drastic differences in

the results for aerosol and fomite exposed animals at 1 DPI, we think that any effect of potential fomite contamination would be negligible.

Can the authors also highlight if severity of disease was related to potency of immunity in the hamsters?

We have now included an analysis of the severity of diseases in relation to humoral immunity in Fig. 5e.

Reference

Rosenke, K., Meade-White, K., Letko, M., Clancy, C., Hansen, F., Liu, Y., Okumura, A., Tang-Huau, T. L., Li, R., Saturday, G., Feldmann, F., Scott, D., Wang, Z., Munster, V., Jarvis, M. A., & Feldmann, H. (2020). Defining the Syrian hamster as a highly susceptible preclinical model for SARS-CoV-2 infection. *bioRxiv : the preprint server for biology*, 2020.09.25.314070. <https://doi.org/10.1101/2020.09.25.314070>

Reviewers' Comments:

Reviewer #1:

Remarks to the Author:

The authors have responded to my concerns. I am satisfied with their responses.

Reviewer #2:

Remarks to the Author:

The authors clearly and thoroughly addressed the points raised in the first review. The manuscript has been significantly improved and I do not have further concerns.

Reviewed by: Nir Paran

Reviewer #3:

Remarks to the Author:

I confirm that the authors have adequately addressed my concerns.

Thank you